# Structural basis of actin filament assembly and aging

Wout Oosterheert[1], Björn U. Klink[1,2], Alexander Belyy[1], Sabrina Pospich[1] & Stefan Raunser[1✉]

The dynamic turnover of actin filaments (F-actin) controls cellular motility in eukaryotes and is coupled to changes in the F-actin nucleotide state[1–3]. It remains unclear how F-actin hydrolyses ATP and subsequently undergoes subtle conformational rearrangements that ultimately lead to filament depolymerization by actin-binding proteins. Here we present cryo-electron microscopy structures of F-actin in all nucleotide states, polymerized in the presence of $Mg^{2+}$ or $Ca^{2+}$ at approximately 2.2 Å resolution. The structures show that actin polymerization induces the relocation of water molecules in the nucleotide-binding pocket, activating one of them for the nucleophilic attack of ATP. Unexpectedly, the back door for the subsequent release of inorganic phosphate ($P_i$) is closed in all structures, indicating that $P_i$ release occurs transiently. The small changes in the nucleotide-binding pocket after ATP hydrolysis and $P_i$ release are sensed by a key amino acid, amplified and transmitted to the filament periphery. Furthermore, differences in the positions of water molecules in the nucleotide-binding pocket explain why $Ca^{2+}$-actin shows slower polymerization rates than $Mg^{2+}$-actin. Our work elucidates the solvent-driven rearrangements that govern actin filament assembly and aging and lays the foundation for the rational design of drugs and small molecules for imaging and therapeutic applications.

Many processes driven by actin, such as cell division, depend on its ATPase activity[1]. In its monomeric form (G-actin), actin exhibits very weak ATPase activity ($7 \times 10^{-6}$ $s^{-1}$) (ref. [4]) but polymerization into filaments (F-actin) triggers a conformational rearrangement that allows actin to hydrolyse ATP within seconds (0.3 $s^{-1}$) (ref. [5]). The cleaved inorganic phosphate ($P_i$) is not released immediately after hydrolysis (release rate 0.006 $s^{-1}$) (ref. [6]), yielding the intermediate ADP-$P_i$ state of F-actin[7]. After the exit of $P_i$, ADP-bound F-actin represents the 'aged' state of the filament, which can then be depolymerized back to G-actin. In vivo, this cyclic process is tightly regulated by various actin-binding proteins (ABPs), of which a subset is capable of sensing the actin nucleotide state[8,9]. As a prominent example, ABPs of the ADF/cofilin family efficiently bind and sever ADP-F-actin to promote actin turnover but only bind with weak affinity to 'young' actin filaments in the ATP or ADP-$P_i$ state[10–12].

In addition to ABPs, the divalent cation that associates with the actin-bound nucleotide, $Mg^{2+}$ or $Ca^{2+}$, also strongly affects polymerization rates. It is now accepted that $Mg^{2+}$ is the predominant cation bound to actin in vivo[13,14]. However, because $Ca^{2+}$-ATP-bound G-actin exhibits slower polymerization kinetics and a higher critical concentration of polymerization[15–17], it has been used as standard in actin purifications[18], many in vitro studies and most G-actin crystal structures[19]. What causes the slow polymerization rates of $Ca^{2+}$-actin remains unknown.

Since 2015, numerous cryo-EM studies have shown the F-actin architecture in all nucleotide states[20–22] and in complex with a variety of ABPs such as cofilin[23,24] and myosin[25–27]. However, previously published F-actin structures were solved at moderate resolutions of about 3–4.5 Å and therefore did not show sufficient details to model solvent molecules and exact positions of amino-acid side-chains. Hence, key mechanistic events in F-actin aging, such as ATP hydrolysis, which strongly depends on water molecules, remain unknown. Here we present 6 cryo-electron microscopy (cryo-EM) structures of rabbit skeletal α-actin filaments at approximately 2.2 Å resolution in 3 functional states, polymerized in the presence of $Mg^{2+}$ or $Ca^{2+}$. The structures illuminate the F-actin architecture in unprecedented detail and underpin the critical role of solvent molecules in actin filament assembly and aging.

## Structures of F-actin reveal solvents

First, by using an optimized cryo-EM workflow (Extended Data Fig. 1 and Methods), we determined structures of $Mg^{2+}$-F-actin in three relevant nucleotide states (ATP, ADP-$P_i$ and ADP) at resolutions of 2.17–2.24 Å (Fig. 1a–d, Extended Data Figs. 2, 3a–f and 4a–c and Supplementary Table 1; Methods).

The unprecedentedly high resolutions of the F-actin reconstructions allowed for the modelling of hundreds of solvent molecules and we accordingly observed clear densities for the nucleotide and the associated $Mg^{2+}$ ion with its coordinating water molecules (Fig. 1b–d and Supplementary Video 1). The overall conformations of all $Mg^{2+}$-F-actin structures are highly similar, with a Cα atom root-mean square deviation of <0.6 Å and no changes in helical rise and twist (Supplementary Table 1 and Extended Data Fig. 4g), indicating that the differences are in the details (see below). Although earlier studies predicted extra $Mg^{2+}$

[1]Department of Structural Biochemistry, Max Planck Institute of Molecular Physiology, Dortmund, Germany. [2]Present address: Centre for Soft Nanoscience, Institute for Medical Physics and Biophysics, University of Münster, Münster, Germany. ✉e-mail: stefan.raunser@mpi-dortmund.mpg.de

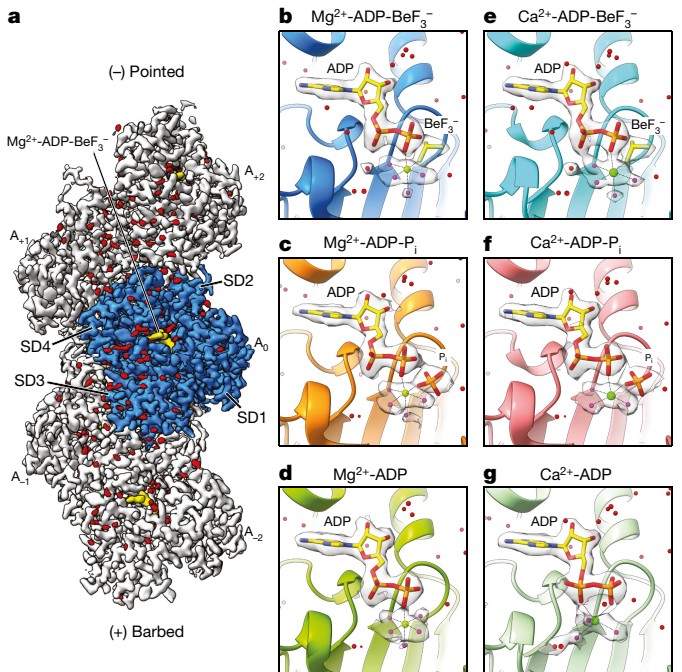

**Fig. 1 | Cryo-EM reconstructions of F-actin at 2.2 Å resolution. a**, Local-resolution filtered, sharpened cryo-EM density map of F-actin in the $Mg^{2+}$-ADP-$BeF_3^-$ state. The subunits are labelled on the basis of their location along the filament, ranging from the barbed ($A_{-2}$) to the pointed ($A_2$) end. The central actin subunit ($A_0$) is blue and the other four subunits are grey. Actin subdomains (SD1–4, also known as Ia, Ib, IIa and IIb) are annotated in the central subunit. Densities corresponding to water molecules are red. **b–g**, Cryo-EM densities of the nucleotide-binding pocket in F-actin in the $Mg^{2+}$-ADP-$BeF_3^-$ (**b**), $Mg^{2+}$-ADP-$P_i$ (**c**), $Mg^{2+}$-ADP (**d**), $Ca^{2+}$-ADP-$BeF_3^-$ (**e**), $Ca^{2+}$-ADP-$P_i$ (**f**) and $Ca^{2+}$-ADP (**g**) states. $Mg^{2+}$ and $Ca^{2+}$ are shown as green spheres. Water molecules that directly coordinate the nucleotide-associated cation are magenta. For the $Ca^{2+}$-ADP structure (**g**), one coordinating water is hidden behind the $Ca^{2+}$ ion.

and $P_i$ binding sites outside of the F-actin nucleotide-binding pocket[19,28], we did not find evidence for these secondary ion-binding sites in any of our reconstructions.

We solved F-actin structures using ADP complexed with beryllium fluoride ($BeF_3^-$, also referred to as $BeF_x$)[29] to mimic the short-lived ATP state of the filament (Fig. 1a,b). In the $Mg^{2+}$-ADP-$BeF_3^-$ F-actin structure (2.17 Å), we observed unambiguous density for the modelling of ADP-$BeF_3^-$ in the nucleotide-binding site (Fig. 1b). Notably, the nucleotide conformation of $Mg^{2+}$-ADP-$BeF_3^-$ in F-actin resembled $Mg^{2+}$-ATP in G-actin (Fig. 2b,c and Extended Data Fig. 5a), confirming the suitability of ADP-$BeF_3^-$ as ATP mimic.

To elucidate the mechanistic basis for the slower polymerization kinetics of $Ca^{2+}$-actin, we solved $Ca^{2+}$-F-actin structures in complex with ADP-$BeF_3^-$, ADP-$P_i$ and ADP at resolutions of 2.15–2.21 Å (Fig. 1e–g, Extended Data Figs. 2, 3g–l and 4d–f, Supplementary Table 2 and Supplementary Video 2). The reconstructions showed that, even though $Ca^{2+}$-actin displays slow polymerization and fast depolymerization kinetics[17], it adopts stable conformations in the filamentous state. Globally, the $Ca^{2+}$-F-actin structures are comparable to those of $Mg^{2+}$-F-actin, with no changes in helical rise and twist and a Cα atom root-mean square deviation of <0.6 Å (Extended Data Fig. 4h–k), indicating that the change from $Mg^{2+}$ to $Ca^{2+}$ does not induce large conformational rearrangements in the filament.

## Water relocation triggers ATP hydrolysis

Upon polymerization, subdomains 1 and 2 (SD1 and SD2) of the actin monomer rotate about 12.4°, leading to a more compact arrangement

in the filament (Fig. 2a and Extended Data Fig. 6a,b), commonly referred to as flattening[30]. A comparison of crystal structures of G-actin in the ATP state[31] with our F-actin structures allows for a description of the G- to F-actin transition in the context of solvent molecules. We first analysed the water molecules directly bound to the nucleotide cation. In $Mg^{2+}$-ADP-$BeF_3^-$ F-actin, $Mg^{2+}$ is coordinated by $P_\beta$ of ADP, a fluoride moiety of $BeF_3^-$ and four water molecules, defining a hexa-coordinated, octahedral coordination, similar to that of $Mg^{2+}$ in ATP-G-actin (Figs. 1b and 2b,c). Our F-actin structure thus provides experimental evidence that $Mg^{2+}$ retains its water coordination during the G- to F-actin transition, which was previously only predicted on the basis of molecular dynamics data[19]. We next inspected the potential relocation of water molecules near the nucleotide-binding pocket. In $Mg^{2+}$-ATP-G-actin (PDB 2V52)[31], there is a large cavity (about 7 Å in diameter) that accommodates several ordered water molecules in front of the ATP γ-phosphate between SD3 and SD1 (SD3/1 cavity, Fig. 2f). Actin flattening results in the upward displacement of the H-loop (residues 72–77) and the movement of the proline-rich loop (residues 108–112) and the side-chains of Q137 and H161 towards the nucleotide (Fig. 2f). As a result, the SD3/1 cavity becomes narrower (about 5 Å in diameter) and a cavity in the SD1 (deemed SD1 cavity) opens up (Extended Data Fig. 6a). Because of the narrowing of the SD3/1 cavity, several water molecules would clash with amino acids and therefore need to relocate to the SD1 cavity through a path which involves the movement of a water molecule ($W_x$) that is bound in between both cavities (Fig. 2f and Supplementary Video 3). Notably, the relocation of water molecules into the SD1 cavity does not impact those coordinating the nucleotide-bound $Mg^{2+}$ ion (Fig. 2c,f).

After the conformational change from G- to F-actin, only three water molecules remain in the SD3/1 cavity that are not coordinated by $Mg^{2+}$. One of them is hydrogen-bonded to the side-chain of Q137 (Fig. 3a and Extended Data Fig. 6c). Owing to the rearrangement of the nucleotide-binding site in F-actin, this water molecule is much closer (3.6 Å) to the Pγ-analogue $BeF_3^-$ than in $Mg^{2+}$-ATP-G-actin (>4 Å distance from the Pγ and 4.6 Å in PDB 2V52; ref. [31]) (Fig. 3a and Extended Data Fig. 6c). As no other ordered water molecules align in front of the nucleotide, the water molecule that is hydrogen-bonded to Q137 is likely to represent the nucleophile ($W_{nuc}$) that hydrolyses ATP in F-actin. The O–Be–$W_{nuc}$ angle in the structure is 144° (Fig. 3a and Extended Data Fig. 6c,i), whereas an angle of >150° is required for efficient nucleophilic attack[19]. Although it cannot be excluded that nucleotide orientation is slightly altered between ADP-$BeF_3^-$-bound and ATP-bound F-actin, inspection of the reconstruction showed that the density for $W_{nuc}$ is extended (Extended Data Fig. 6e), indicating that the position of $W_{nuc}$ is not fixed, allowing it to move into a position that brings the O–Be–$W_{nuc}$ angle >150° while remaining hydrogen-bonded to Q137. In other words, $W_{nuc}$ probably exchanges between hydrolysis-competent and hydrolysis-less-competent configurations.

Although Q137 positions $W_{nuc}$ in close proximity to Pγ, the Q137 side-chain cannot accept a proton to act as a catalytic base for the hydrolysis. We found no other amino acids that are close enough to interact with $W_{nuc}$. Instead, $W_{nuc}$ resides at about 4.2 Å from a neighbouring water molecule ($W_{bridge}$) (Fig. 3a), which is not close enough to form a hydrogen bond but the movement of $W_{nuc}$ into a hydrolysis-competent position would also place $W_{nuc}$ in hydrogen-bonding distance to $W_{bridge}$. By forming hydrogen bonds with D154 and H161, $W_{bridge}$ may represent a Lewis base with a high potential to activate $W_{nuc}$ and potentially act as an initial proton acceptor during hydrolysis, followed by transfer of the proton to D154, as previously predicted by simulations[32,33] or, alternatively, to H161. In conclusion, we propose that Q137 coordinates $W_{nuc}$ but that the hydrogen-bond network comprising $W_{bridge}$, D154 and H161 is responsible for the activation of $W_{nuc}$ and proton transfer. Indeed, the ATP hydrolysis rates of the Q137 to alanine (Q137A) actin mutant are slower but not abolished[34], whereas the triple mutant Q137A/D154A/H161A-actin exhibits no measurable ATPase activity[35].

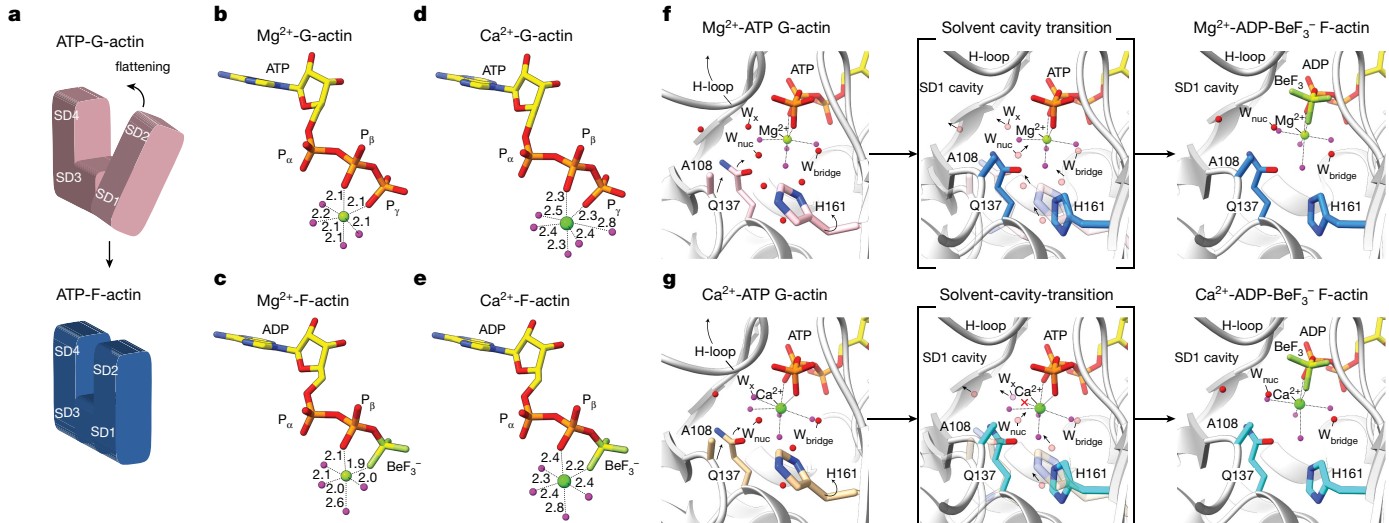

**Fig. 2 | Water relocation during the G- to F-actin transition. a,** Schematic cartoon representation of actin flattening during the G- to F-actin transition. **b–e,** Nucleotide conformation and inner-coordination sphere of the divalent cation in $Mg^{2+}$-ATP-G-actin (Protein Data Bank (PDB) 2V52) (**b**), $Mg^{2+}$-ADP-$BeF_3^-$ F-actin (**c**), $Ca^{2+}$-ATP-G-actin (PDB 1QZ5) (**d**) and $Ca^{2+}$-ADP-$BeF_3^-$ F-actin (**e**). Bond lengths are annotated in angstroms. **f,g,** Water relocation in $Mg^{2+}$-actin (**f**) and $Ca^{2+}$-actin (**g**). In **f** and **g**, the left panel shows water and amino-acid arrangement in ATP-G-actin. Amino acids are pink for $Mg^{2+}$-actin and light-brown for $Ca^{2+}$-actin, whereas the cartoon representation is shown in grey. Arrows depict the movement of amino-acid regions for the transition to F-actin. The middle panel shows overlay of the amino-acid positions in ADP-$BeF_3^-$ F-actin (blue for $Mg^{2+}$-actin and cyan for $Ca^{2+}$-actin) with the solvent molecules in the G-actin structure. The water molecules in the SD3/1 cavity of ATP-G-actin are shown as semitransparent spheres. Arrows indicate the direction of water relocation. Finally, the right panel shows the water and amino-acid arrangement in ADP-$BeF_3^-$ F-actin. Nuc, nucleophilic.

## Slow polymerization of $Ca^{2+}$-actin

We next inspected the G- to F-actin transition in $Ca^{2+}$-actin structures. The $Ca^{2+}$ ion in G-actin is coordinated by the $P_\beta$ and $P\gamma$ of ATP and five water molecules in a hepta-coordinated, pentagonal-bipyramidal arrangement (Fig. 2d)[36,37]. By contrast, in $Ca^{2+}$-ADP-$BeF_3^-$ F-actin, the $Ca^{2+}$ ion loses one coordinating water and displays an octahedral coordination sphere (Fig. 2e). How does the G- to F-actin transition lead to changes in $Ca^{2+}$-coordination? Globally, the flattening of $Ca^{2+}$-actin triggers rearrangements that are analogous to those observed in $Mg^{2+}$-actin (Extended Data Fig. 6a,b), with a similar relocation of ordered water molecules from the narrowing SD3/1 cavity to the widening SD1 cavity (Fig. 2g). However, in $Ca^{2+}$-G-actin, one of the relocating water molecules ($W_x$) resides within the coordination sphere of $Ca^{2+}$, indicating that the hydration shell of the $Ca^{2+}$ ion needs to be altered for the G- to F-actin transition to occur. Thus, our analysis rationalizes why the inner-sphere coordination of $Ca^{2+}$ changes from hepta-coordinated, pentagonal-bipyramidal in ATP-G-actin to hexa-coordinated, octahedral in ADP-$BeF_3^-$ F-actin (Fig. 2d,e,g and Extended Data Fig. 5b). The required rearrangement of the $Ca^{2+}$-coordination sphere could pose a kinetic barrier for the G- to F-actin transition, which provides a structural basis for the slower polymerization kinetics of $Ca^{2+}$-actin compared to $Mg^{2+}$-actin.

We also assessed the ATP hydrolysis mechanism of $Ca^{2+}$-actin, which exhibits a five times slower ATP hydrolysis rate ($0.06\ s^{-1}$) than $Mg^{2+}$-actin ($0.3\ s^{-1}$) (ref.[5]). A structural comparison between $Ca^{2+}$-ATP-G-actin (PDB 1QZ5) and $Ca^{2+}$-ADP-$BeF_3^-$ F-actin demonstrates that the water corresponding to $W_{nuc}$, which is hydrogen-bonded to Q137, locates closer to Be in F-actin (3.7 Å) than to $P\gamma$ in G-actin (4.5 Å) (Extended Data Fig. 6d,h,i). Thus, the induction of ATP hydrolysis is comparable in $Ca^{2+}$-actin and $Mg^{2+}$-actin. However, in $Ca^{2+}$-ADP-$BeF_3^-$ F-actin, the distance between Q137 and $W_{nuc}$ is 3.4 Å (3.2 Å in $Mg^{2+}$-actin) and the $O-Be-W_{nuc}$ angle is 137° (144° in $Mg^{2+}$-actin) (Extended Data Fig. 6c–i), making the position of $W_{nuc}$ similar, but slightly less favourable for nucleophilic attack, providing a likely explanation for the slower hydrolysis rate of $Ca^{2+}$-F-actin.

## $P_i$ release occurs in a transient state

We next analysed how ATP hydrolysis affects the F-actin nucleotide arrangement. In the $Mg^{2+}$-ADP-$P_i$ state, the cleaved $P_i$ moiety is separated from ADP by at least 2.9 Å (Figs. 1c and 3b and Extended Data Fig. 5a), indicating that ADP and $P_i$ do not form a covalent bond. After $P_i$ release, the $P_i$-binding site is occupied by a water molecule, which also holds true for structures of monomeric $Mg^{2+}$-ADP-G-actin[38] (Extended Data Fig. 5a). Taken together, our structures show that the $Mg^{2+}$-coordination shell is octahedral and that $Mg^{2+}$ resides at a fixed position beneath the $P_\beta$ moiety of the nucleotide in all stable states of G-actin and F-actin.

After ATP hydrolysis in $Ca^{2+}$-F-actin, the coordination of $Ca^{2+}$ is also octahedral in the ADP-$P_i$ state (Fig. 1f and Extended Data Figs. 5b and 6h). Interestingly, one coordinating water molecule is replaced by the side-chain of Q137 (Extended Data Fig. 5c–e). Finally, following $P_i$ release, the $Ca^{2+}$-ADP-F-actin structure shows that the $Ca^{2+}$ ion changes position so that it is directly coordinated by both the $P_\alpha$ and $P_\beta$ of ADP (Fig. 1g and Extended Data Fig. 5b) and four water molecules in an octahedral arrangement. Thus, in contrast to $Mg^{2+}$, the $Ca^{2+}$ ion position is not fixed in F-actin and its coordination changes considerably during the ATPase cycle. The absence of discrete differences in amino-acid conformation between the ADP states of $Ca^{2+}$- and $Mg^{2+}$-F-actin suggests that the faster depolymerization rates of $Ca^{2+}$-F-actin may be caused by differences in long-range filament mechanostability or conformations at filament ends.

$P_i$ is thought to exit from the F-actin interior through the so called 'back door'[39], which is formed by the side-chains of R177 and N111 and the backbones of methylated histidine 73 (H73) and G74 (Fig. 3c,d and Extended Data Fig. 5f,g). In this model, S14 switches rotameric position to change its hydrogen-bonding interaction from the backbone amide of G74 to the one G158, thereby allowing $P_i$ to approach the back door, where R177 would mediate its exit[22,39]. On the basis of a lower-resolution reconstruction, the back door was proposed to be open in ADP-F-actin[22]. However, the S14–G74 hydrogen bond is intact and the back door is closed in our 2.2 Å structures of both the ADP-$P_i$ and ADP states of $Mg^{2+}$-F-actin (Fig. 3c,d) and $Ca^{2+}$-F-actin (Extended Data Fig. 5f,g).

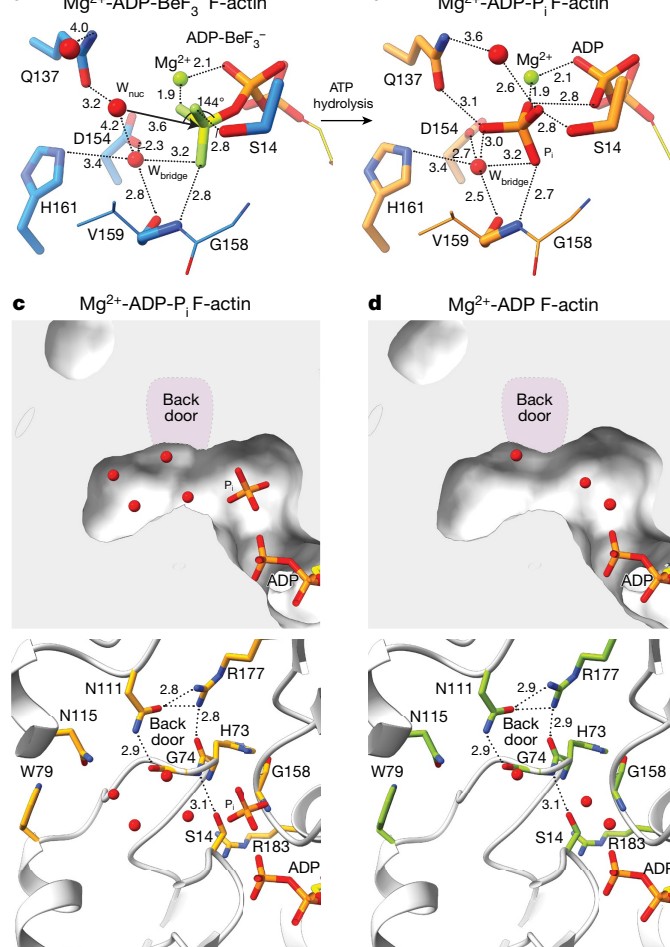

**a** Mg²⁺-ADP-BeF₃⁻ F-actin

**b** Mg²⁺-ADP-Pᵢ F-actin

ATP hydrolysis →

**c** Mg²⁺-ADP-Pᵢ F-actin

**d** Mg²⁺-ADP F-actin

Back door

**Fig. 3 | Structural insights into ATP hydrolysis and Pᵢ release. a,b,** Isolated amino-acid and water arrangement near the nucleotide in Mg²⁺-ADP-BeF₃⁻ F-actin (**a**) and Mg²⁺-ADP-Pᵢ F-actin (**b**). Regions unimportant for interactions are depicted as smaller sticks. Amino acids and the proposed nucleophilic water ($W_{nuc}$) and assisting water ($W_{bridge}$) are annotated. **c,d,** Internal solvent cavities near the Pᵢ binding site in ADP-Pᵢ (**c**) and ADP (**d**) structures of Mg²⁺-F-actin. The upper panel shows the F-actin structure as surface with the bound Pᵢ and water molecules. In the lower panel, F-actin is shown in cartoon representation and the amino acids forming the internal cavity are annotated and shown as sticks. Hydrogen bonds are depicted as dashed lines. The position of the proposed back door is highlighted in purple in the upper panel. All distances are shown in angstroms.

Thus, unexpectedly, our structures show that the back door closes again after Pᵢ release, indicating that the F-actin conformation that allows for the exit of Pᵢ is a transient state. In fact, the proposed rotameric switch of S14 towards G158 alone would not result in an opened back door, which suggests that larger rearrangements are required for Pᵢ release, indicating that the release mechanism remains incompletely understood. We envision that Pᵢ release could be further explored by molecular dynamics simulations or time-resolved cryo-EM in future research, guided by our structures as high-quality starting models.

## Coupling of filament centre to periphery

We next examined the nucleotide state-dependent conformational mobility of the D-loop (residues 39–51) and the carboxy terminus at the intrastrand (or longitudinal) interface in the actin filament[21,40]. The intrastrand arrangements in the current high-resolution Mg²⁺-F-actin structures are largely consistent with those in previous

reconstructions[21], with a mixture of open/closed D-loop conformations in 'young' ATP-bound filaments and a predominantly closed D-loop arrangement in 'aged' ADP-F-actin (Extended Data Fig. 7). In the Mg²⁺-ADP-BeF₃⁻ F-actin structure, we could separate two intrastrand conformations through a focused classification approach (Extended Data Fig. 8a,b). The first conformation (about 37% of the particles, 2.32 Å resolution) represents the open D-loop, where the D-loop bends outwardly and interacts with the extended C terminus of the adjacent actin subunit. In the second conformation (about 63% of the particles, 2.32 Å resolution), the C terminus remains extended but turns away from the inwardly folded, closed D-loop (Extended Data Fig. 7). In Mg²⁺-ADP-Pᵢ F-actin, the C terminus forms a compact, folded α-helix and the D-loop is predominantly closed, whereas the Mg²⁺-ADP-F-actin structure resembles the extended C terminus and closed D-loop conformation of the Mg²⁺-ADP-BeF₃⁻ state (Extended Data Fig. 7).

We next analysed how conformational changes in the nucleotide-binding pocket are transmitted to the filament surface. Surprisingly, we could not identify a direct communication path between the D-loop and the nucleotide-binding site (Extended Data Fig. 8c–e). Hence, our structures do not explain why the intrastrand interface can adopt two conformations in the ATP state of Mg²⁺-actin. However, we were able to identify the structural basis for the nucleotide-dependent conformation of the C terminus. After ATP hydrolysis, Q137 in the nucleotide-binding pocket moves upward by about 0.4 Å in the ADP-Pᵢ state so that it resides within 3.1 Å of Pᵢ (Figs. 3b and 4 and Supplementary Video 4). This upward movement of Q137 triggers a sequence of small movements in the SD1; the proline-rich loop (residues 108–112) moves slightly forward and triggers the relocation of the E107–R116 salt bridge, which allows the penultimate residue C374 to flip into a hydrophobic pocket, permitting R116 to interact with the carboxylate group of the C-terminal residue F375 (Fig. 4 and Supplementary Video 4). Altogether, these changes result in a compact, folded C-terminal helix, which then unfolds again when Q137 moves downward after Pᵢ release (Extended Data Fig. 9). In conclusion, our data suggest that Q137 and its surrounding residues represent a major region that is capable of sensing the nucleotide state and transmitting it to the periphery.

## Nucleotide-state sensing by cofilin

It has been proposed that the intrastrand interface represents a major site for ABPs such as cofilin to sense the nucleotide state of F-actin. Cofilin binds and changes the helical twist of actin filaments by wedging itself between the C terminus and D-loop[23,24,41,42], which may be inhibited by the open D-loop conformation[21]. In the structures of Ca²⁺-F-actin, we observed similar arrangements of the intrastrand interface compared to those of Mg²⁺-F-actin, except that the open D-loop conformation is adopted to a lesser extent in Ca²⁺-ADP-BeF₃⁻ F-actin (Extended Data Fig. 7). We therefore proposed that if the D-loop arrangement represents the dominant recognition signal, cofilin would efficiently bind and sever Ca²⁺-F-actin regardless of the nucleotide state. To assess this, we incubated cofilin-1 and Mg²⁺- or Ca²⁺-bound F-actin in three nucleotide states and measured cofilin-dependent filament severing. The assays showed that, comparable to Mg²⁺-F-actin, cofilin-1 only substantially severs the ADP state but not to ADP-BeF₃⁻ and ADP-Pᵢ-bound Ca²⁺-F-actin (Extended Data Fig. 10a). Therefore, the D-loop conformation does not represent the only sensor for cofilin-1 binding. What other mechanism does cofilin use to sense the nucleotide state? Our structures show that BeF₃⁻ and Pᵢ make hydrogen-bonding interactions with S14 of SD1; and the backbones of G158 and V159 of the SD3 (Extended Data Fig. 10c–h). Thus, the γ-phosphate moiety forms a bridge between the two subdomains, which is absent in the ADP state. Previous studies indicated that the tight binding of cofilin to F-actin necessitates a change in helical twist of the filament[24], which involves the rotation of SD1 and 2 (ref. [42]; Extended Data Fig. 10b). This rotation involves the movement of the loop of S14, which is not possible when S14 is hydrogen-bonded to the

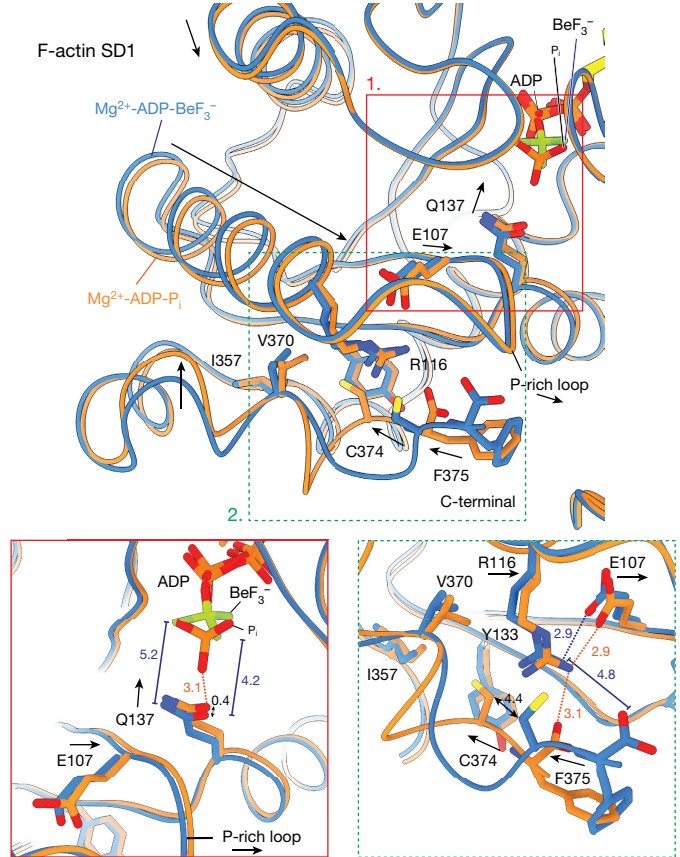

**Fig. 4 | Structural coupling of the nucleotide-binding site to the filament surface.** Top: differences in the SD1 of F-actin in the Mg$^{2+}$-ADP-BeF$_3^-$ and Mg$^{2+}$-ADP-P$_i$ structures. Residues thought to be important for the movement are annotated. Bottom: zoom of the nucleotide-binding site (1) and C-terminal region (2) of the SD1. Arrows depict the direction of the putative movement from the Mg$^{2+}$-ADP-BeF$_3^-$ to the Mg-ADP-P$_i$ structure. All distances are shown in angstroms. Distances shown in the Mg$^{2+}$-ADP-BeF$_3^-$ structure are blue, whereas those in the Mg-ADP-P$_i$ structure are orange.

γ-phosphate of the nucleotide in ATP or ADP-P$_i$-F-actin (Extended Data Fig. 10c–i). Our results therefore support the previously proposed model that cofilin cannot form a strong complex with F-actin when the γ-phosphate moiety is present[23] and that it potentially senses the mechanical properties of the filament. This is in agreement with numerous biochemical observations that ADP/cofilin proteins can only sever actin filaments when P$_i$ or BeF$_3^-$ is removed from the active site[10,11,43].

## Conclusions

The structures of F-actin at about 2.2 Å resolution show the filament architecture and the arrangement of the nucleotide-binding pocket in unmatched detail, allowing us to revise certain statements about the flexibility and stability of F-actin. Traditionally, the structure of F-actin has been described as polymorphic[44], whereas 'aged' ADP-F-actin is regarded as a structurally destabilized form of the filament[45]. By contrast, our structures are remarkably similar in all solved nucleotide states, showing that ADP-F-actin should not be regarded as destabilized but rather as a 'primed state', which exhibits faster depolymerization rates at the filament ends and is sensitive to cofilin binding and severing due to the absence of the γ-phosphate moiety. This model is highly consistent with a recent study, which showed that the nucleotide state affects the bending and mechanical properties of the filament, rather than large amino-acid rearrangements[46]. We furthermore show how the mechanism of ATP hydrolysis in F-actin and the slow polymerization

rates of Ca$^{2+}$-actin depend on the positions of water molecules, emphasizing that high-resolution structures are crucial for explaining these important aspects of filament assembly and aging. Our optimized cryo-EM workflow now also paves the way for high-resolution structures of F-actin bound to ABPs, which will enhance our understanding of cytoskeletal remodelling. Finally, we envision that our solvent-molecule visualizing structures of F-actin may serve as high-quality templates for the development of actin-binding small molecules, which may be tailored for imaging and, perhaps, even therapeutic applications[47,48].

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

## Methods

### Protein purification

Skeletal α-actin was purified from rabbit muscle acetone powder through an established protocol that was described previously[18,21,49]. A total of 0.5 g of frozen muscle acetone powder was thawed, resuspended in 10 ml of G-buffer (5 mM Tris pH 7.5, 0.2 mM CaCl₂, 0.2 mM ATP, 0.5 mM tris(2-carboxyethyl)phosphine (TCEP), 0.1 mM NaN₃) and stirred for 25 min at 4 °C. The suspension was then filtered and the pellet was again resuspended in 10 ml of G-buffer and subjected to the same stirring procedure. After filtering, the 20 ml of filtered solution was ultracentrifuged at 100,000$g$ for 30 min to remove any remaining debris. The supernatant was collected and actin was polymerized by the addition of 2 mM MgCl₂ and 100 mM KCl (final concentrations) for 1 h at room temperature. To remove ABPs bound to actin, solid KCl was added to the solution to bring the KCl concentration to 800 mM and the mixture was incubated for 1 h at room temperature. Then, the actin filaments were pelleted by ultracentrifugation at 100,000$g$ for 2 h and resuspended in 5 ml of G-buffer. Actin was depolymerized by dialysis in 1 l of G-buffer for 2 d, with one buffer exchange per day. This ensured that Ca²⁺ was the divalent cation bound in the active site of G-actin. On the third day, the solution was ultracentrifuged at 100,000$g$ for 30 min and G-actin was recovered from the supernatant. The 2 d procedure of actin polymerization, high-salt wash and depolymerization by dialysis in G-buffer was repeated once more to ensure removal of all impurities and ABPs. After depolymerization, purified G-actin was flash frozen in liquid nitrogen in 50 µl aliquots at a concentration of 28 µM and stored at −80 °C until further use.

Human cofilin-1 was purified as described previously[50].

### Reconstitution of F-actin in different functional states

Structural studies were performed on rabbit skeletal α-actin, which is identical to human skeletal α-actin in amino-acid sequence. G-actin aliquots were thawed and ultracentrifuged for 1 h at 100,000$g$ to remove aggregates. For structures determined with Mg²⁺ as nucleotide-associated cation, G-actin (28 µM) was mixed with 0.5 mM EGTA and 0.2 mM MgCl₂ to exchange Ca²⁺ for Mg²⁺ 5–10 min before polymerization. In all subsequent steps, buffers contained CaCl₂ for the isolation of F-actin with Ca²⁺ as divalent cation or MgCl₂ for the isolation of F-actin with Mg²⁺ as divalent cation. Actin polymerization was induced by the addition of 100 mM KCl and 2 mM CaCl₂/MgCl₂ (final concentrations). Actin was polymerized at room temperature for 2 h and subsequently overnight at 4 °C. The next morning, filaments were isolated through ultracentrifugation at 100,000$g$ for 2 h.

For the aged ADP-F-actin states, the filament pellet was resuspended in F⁻ buffer: 5 mM Tris pH 7.5, 100 mM KCl, 2 mM CaCl₂/MgCl₂, 2 mM NaN₃, 1 mM dithiothreitol (DTT). F-actin was used for cryo-EM sample preparation about 1 h after pellet resuspension.

When choosing an ATP analogue for structural studies, we considered that previous work from our group has shown that the widely used non-hydrolysable ATP analogue AppNHp (also known as AMP-PNP) is a suboptimal ligand for F-actin because its degradation product, the ADP analogue AppNH₂, exhibits higher affinity for F-actin[51] and hence accumulates in the active site during filament preparation[21]. We therefore opted to use ADP-BeF₃⁻ as mimic of ATP. To obtain these ADP-BeF₃⁻ states of F-actin, aged ADP-bound filaments were resuspended in F⁻ buffer supplemented with 0.75 mM BeF₂ and 5 mM NaF. Because the on-rate of BeF₃⁻ for ADP-F-actin is relatively slow[29], the filaments were incubated in this buffer for >6 h before cryo-EM sample preparation to ensure saturation with BeF₃⁻.

To isolate F-actin in the ADP-P_i state, we resuspended the actin pellet in F⁻ phosphate buffer: 5 mM Tris, 50 mM KCl, 2 mM CaCl₂/MgCl₂, 2 mM NaN₃, 1 mM DTT, 50 mM potassium phosphate pH 7.5. To remove any potential precipitates of calcium phosphate and magnesium phosphate, we filtered the buffers directly before use. The filaments were incubated in F⁻ phosphate buffer for >6 h before cryo-EM sample preparation.

### Cryo-EM grid preparation

A total 2.8 µl of F-actin sample (3–16 µM) was pipetted onto a glow-discharged R2/1 Cu 300 mesh holey-carbon grid (Quantifoil). After incubating for 1–2 s, excess solution was blotted away and the grids were plunge frozen in liquid ethane or a liquid ethane/propane mixture using a Vitrobot Mark IV (Thermo Fisher Scientific). The Vitrobot was operated at 13 °C and the samples were blotted for 9 s with a blot force of −25.

### Cryo-EM grid screening and data collection

Grids were prescreened on a 200 kV Talos Arctica Microscope (Thermo Fisher Scientific) equipped with a Falcon III detector (Thermo Fisher Scientific). Typically, low-magnification grid overviews (atlases) were collected using EPU (Thermo Fisher Scientific). Afterwards, around two holes per grid square were imaged at high magnification for a total of five grid squares to visualize F-actin. The grids that displayed optimal filament concentration and distribution were then retrieved from the microscope and stored in auto grid boxes (Thermo Fisher Scientific) in liquid nitrogen until further use for high-resolution data collection.

All datasets were collected on a 300 kV Titan Krios microscope (Thermo Fisher Scientific) equipped with a K3 detector (Gatan) and a postcolumn energy filter (slit width of 15 eV). Videos were obtained in super-resolution mode at a pixel size of 0.3475 Å, with no objective aperture inserted. All datasets were collected on the same microscope at the same magnification of ×130,000, to ensure that the resulting cryo-EM density maps could be compared directly without issues caused by pixel size discrepancies. Using EPU, we collected about 10,000 videos per dataset in 60–80 frames at a total electron exposure of about 72–90 e⁻ Å⁻². The defocus values set in EPU ranged from −0.7 to −2.0 µm. The data quality was monitored live during acquisition using TranSPHIRE[52]. If necessary, the microscope was realigned to ensure optimal imaging conditions. An overview of the collection settings used for each dataset can be found in Supplementary Tables 1 and 2.

### Cryo-EM image processing

For each dataset, video preprocessing was performed on the fly in TranSPHIRE[52], the super-resolution videos were binned twice (resulting pixel size of 0.695 Å), gain corrected and motion corrected using UCSF MotionCor2 (ref. [53]), contrast transfer function (CTF) estimations were performed with CTFFIND4.13 (ref. [54]) and F-actin segments were picked using the filament picking procedure in SPHIRE_crYOLO[55,56] using a box distance of 40 pixels per 27.8 Å and a minimum number of six boxes per filament. The resulting particles were extracted in a 384 × 384 pixel box and further processed into the pipeline of helical SPHIRE v.1.4 (ref. [57]). The number of extracted particles differed per dataset and ranged from 1,296,776 (Mg²⁺-ADP dataset) to 3,031,270 (Ca²⁺-ADP-P_i dataset) particles. For each dataset, the particles were two-dimensionally classified in batches of 20,000 particles using ISAC2 (ref. [58]) (sp_isac2.py). All classes were then pulled together and manually inspected and those that represented non-filament picks and ice contaminations were discarded. A virtual substack was created of the remaining particles (sp_pipe.py isac_substack) and the particles were subjected to three-dimensional helical refinement[52] using meridien alpha. This refinement approach within SPHIRE imposes helical restraints tailored to the helical sample to facilitate the refinement process but does not apply helical symmetry. Hence, symmetrization artifacts during refinement are avoided. We refined the F-actin structures with a restrained tilt angle during exhaustive search (--theta_min 90 −theta_max 90 −howmany 10) and used a filament width of 140 pixels (97.3 Å) and a helical rise of 27.5 Å to limit shifts larger than one subunit to prevent duplication of particles. For the first processed dataset, EMD-11787 (ref. [47]) was lowpass filtered to 25 Å and supplied as initial model for the refinement. The first meridien alpha refinement of each dataset was performed without a mask; the resulting three-dimensional density map of this refinement was then

used to create a soft mask using sp_mask.py that covered about 85% of the filament (326 pixels in the Z-direction). The global refinement was then repeated with the same settings but with the mask applied. These masked refinements yielded F-actin reconstructions at resolutions of 2.6–3.0 Å. The particles were then converted to be compatible with Relion[59] using sp_sphire2relion.py. Within Relion 3.1.0, the particles were subjected to Bayesian polishing[60] for improved estimation of particle movement trajectories caused by beam-induced motion; and to CTF refinements[61] to estimate per-particle defocus values and to correct for beam tilt, threefold (trefoil) astigmatism, Cs and fourfold (tetra-foil) astigmatism and anisotropic magnification. We then performed three-dimensional classification without image alignment (8 classes, 25 iterations, tau2fudge 4) to remove particles that did not contribute high-resolution information to the reconstruction. Typically, one or two high-resolution classes containing most particles were selected and the other low-resolution classes were discarded. Finally, after removal of duplicates, this set of particles was subjected to a masked refinement with solvent flattening Fourier shell correlations (FSCs) and only local searches (initial sampling 0.9°) in Relion using the map (lowpass filtered to 4.0 Å), mask and particle orientations determined from SPHIRE. These refinements yielded cryo-EM density maps at resolutions of 2.15–2.24 Å according to the gold-standard FSC = 0.143 criterion. The final maps were sharpened with a negative B-factor and corrected with the modulation transfer function of the K3 detector. Local-resolution estimations were performed in Relion.

To separate the closed and open D-loop conformations in the $Mg^{2+}$-ADP-$BeF_3^-$ reconstruction, the good 2,228,553 particles of this dataset were subjected to a focused classification without image alignment in Relion. We created a soft mask around an inter-F-actin contact comprising the D-loop of the central actin subunit and the C terminus of the subunit directly above. Initial attempts to separate the D-loop conformations into two classes using a single density map as initial model were unsuccessful because all particles would end up in a single class with a mixed closed/open D-loop population. We therefore classified the particles into two classes using two references; the jasplakinolide-bound, $Mg^{2+}$-ADP-$P_i$ (in-house structure) and $Mg^{2+}$-ADP structures as templates for, respectively, open and closed D-loop conformations. A particle separation into two classes with two initial references was chosen because the two conformations of closed and open D-loops were visible in the refined, non-sharpened reconstruction computed through all 2,228,553 good particles. Potential alternative conformations of the D-loop adopted by only a marginal number of F-actin particles are not distinguishable by our classification strategy. After optimization of the tau2fudge parameter, which required a high value due to the small size of the mask compared to the full box, this classification without image alignment (two classes, 25 iterations, tau2fudge = 500) yielded two classes with clearly distinguishable D-loop conformations. The particles belonging to each class (834,110 for the open D-loop and 1,394,443 for the closed D-loop) were then selected and separately refined in Relion with solvent flattening FSCs using the map with mixed D-loop conformation (filtered to 8.0 Å, at which the D-loop conformation is indistinguishable) as reference and the SPHIRE-mask covering 85% of the filament for both refinements. These refinements were performed with the default tau2fudge = 1 to prevent any overfitting. The high tau2fudge value used during the classification without image alignment was only used to sort particles and not for further map processing and analysis. The resulting maps of the open D-loop (2.32 Å) and closed D-loop (2.32 Å) particles showed, respectively, the expected open and closed D-loop conformations at the region that was used for focused classification. The D-loop conformations remained a mix between open and closed in actin subunits within the map that were not used for focused classification.

## Model building, refinement and analysis

To build the F-actin models in the high-resolution density maps, the structure of F-actin in complex with an optojasp in the *cis* state[47] (PDB 7AHN) was rigid-body fitted into the map of F-actin $Ca^{2+}$-ADP state. We modelled five actin subunits in each map to capture the entire interaction interface within the filament because the central subunit interacts with four neighbouring protomers. The central actin subunit in the map was rebuilt manually in Coot[62] and the other actin subunits were adjusted in Coot by applying non-crystallographic symmetry using the central subunit as master chain. The structure was then iteratively refined using Coot (manually) and phenix real-space refine[63] with non-crystallographic symmetry restraints but without imposing any geometry restraints. The structures of all other states were built by rigid-body fitting of the $Ca^{2+}$-ADP structure in the map belonging to each F-actin state, followed by manual adjustments in Coot. These structures were then refined through a similar protocol of iterative cycles in Coot and phenix real-space refine. All solvent molecules (ions and water molecules) were placed manually in Coot in the central actin subunit and were then placed in the other subunits using non-crystallographic symmetry. Because the local resolution of each F-actin reconstruction is highest in the centre and lower at the periphery of the map, we inspected all water molecules in each structure manually before the final phenix refinement; water molecules with poor corresponding cryo-EM density were removed. A summary of the refinement quality for each structure is provided in Supplementary Tables 1 and 2. For the structural analysis, the central actin subunit in the structure was used, unless stated otherwise. The solvent cavities in the structures were calculated using the CASTp 3.0 web server[64]. All figures that depict cryo-EM density maps and protein structures were prepared in UCSF ChimeraX[65]. The helical parameters reported in Supplementary Tables 1 and 2 were estimated from the atomic model of five consecutive subunits independently fitted to the map as described previously[66].

## Cofilin severing assays

F-actin in different nucleotide states was prepared as for cryo-EM experiments (see above). Severing assays were performed in 20 µl volumes by incubating 5 µM of F-actin with 5, 10 or 20 µM of cofilin for 30 min at room temperature, followed by centrifuging the samples at 120,000*g* in a TLA120.1 rotor for 15 min at 4 °C. After centrifugation, aliquots of the supernatant and pellet fractions were separated by SDS–polyacrylamide gel electrophoresis and analysed by densitometry using Image Lab software v.6.0.1 (Bio-Rad) and plotted using GraphPad. The data points are available as Source data.

## Reporting summary

Further information on research design is available in the Nature Research Reporting Summary linked to this article.

## Data availability

The cryo-EM maps have been deposited to the Electron Microscopy Data Bank under accession codes (dataset in brackets): EMD-15104 ($Mg^{2+}$-ADP-$BeF_3^-$), EMD-15105 ($Mg^{2+}$-ADP-$P_i$), EMD-15106 ($Mg^{2+}$-ADP), EMD-15107 ($Ca^{2+}$-ADP-$BeF_3^-$), EMD-15108 ($Ca^{2+}$-ADP-$P_i$) and EMD-15109 ($Ca^{2+}$-ADP). These depositions include sharpened maps, unfiltered half-maps and the refinement masks. For the $Mg^{2+}$-ADP-$BeF_3^-$ F-actin submission, all density maps and masks regarding the separation of open/closed D-loop conformations are provided. The atomic coordinates of the protein structures have been submitted to the Protein Data Bank under accession codes (dataset in brackets): 8A2R ($Mg^{2+}$-ADP-$BeF_3^-$), 8A2S ($Mg^{2+}$-ADP-$P_i$), 8A2T ($Mg^{2+}$-ADP), 8A2U ($Ca^{2+}$-ADP-$BeF_3^-$), 8A2Y ($Ca^{2+}$-ADP-$P_i$) and 8A2Z ($Ca^{2+}$-ADP). We used the following previously published structures for modelling and comparisons: 7AHN, 2V52, 6RSW, 1QZ5 and 1J6Z. EMD-11787 was used as the initial model for the first 3D refinement. Source data are provided with this paper.

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

**Acknowledgements** We gratefully thank D. Prumbaum and O. Hofnagel for the assistance with cryo-EM data collection. We also thank S. Bergbrede for technical support in the wet laboratory and T.D. Pollard, F. Merino and R.S. Goody for critical proofreading of the manuscript. We acknowledge W. Linke and A. Unger for supplying us with muscle acetone powder. This work was supported by funds from the Max Planck Society (to S.R.) and the European Research Council under the European Union's Horizon 2020 Programme (ERC-2019-SyG, grant no. 856118 to S.R). A.B. is supported by an EMBO long-term fellowship. W.O. is supported by a postdoctoral fellowship from the Alexander von Humboldt foundation.

**Author contributions** S.R. conceived and supervised the study. B.U.K. and S.P. optimized data collection strategies. W.O. collected and processed all cryo-EM data and built the atomic models. W.O., S.P. and S.R. analysed the structures. A.B. performed the cofilin severing assays. W.O. and S.R. wrote the manuscript, with critical input from all authors.

**Funding** Open access funding provided by Max Planck Society.

**Competing interests** The authors declare no competing interests.

**Additional information**
**Correspondence and requests for materials** should be addressed to Stefan Raunser.

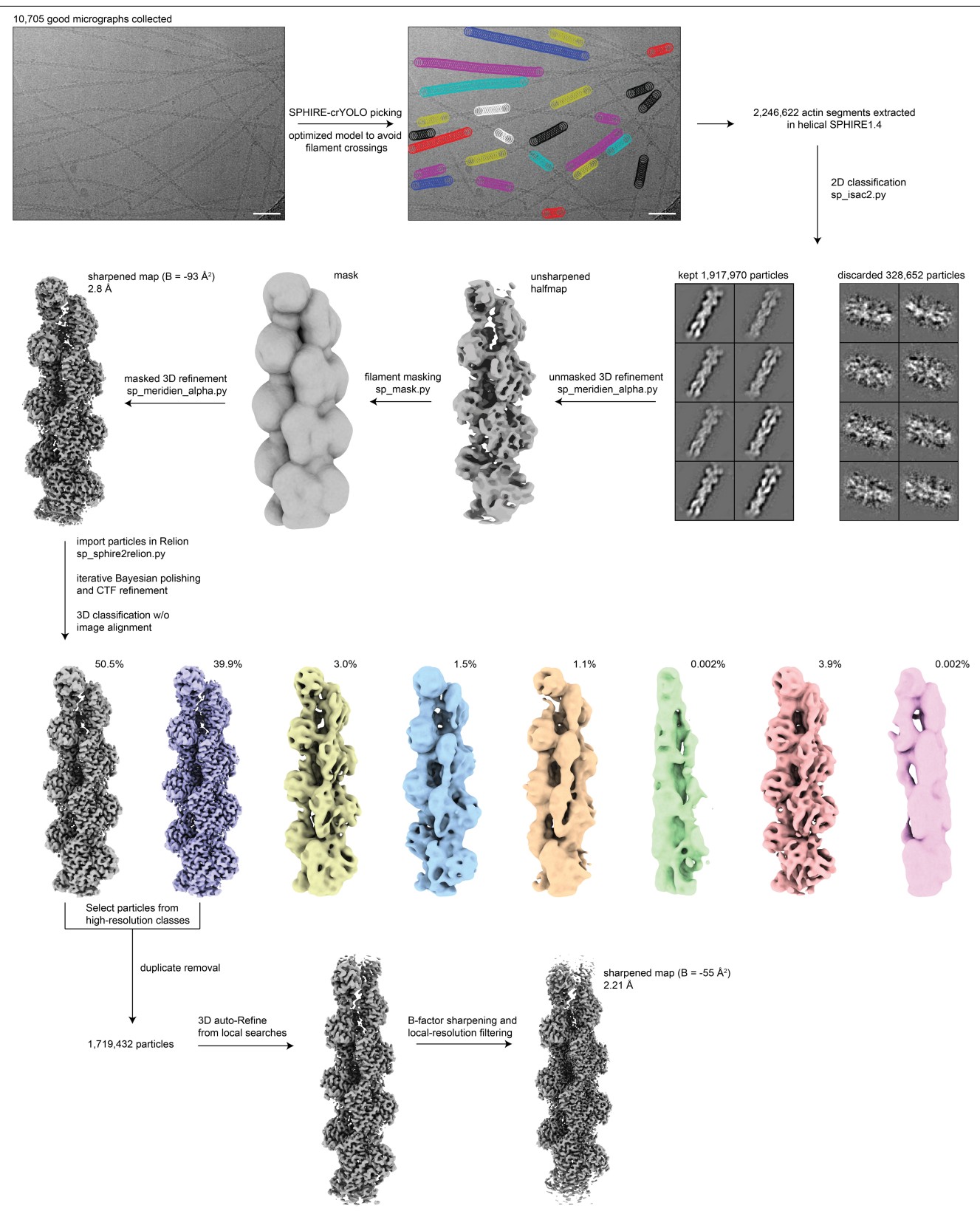

**Extended Data Fig. 1 | Cryo-EM image processing workflow.** The image processing workflow that was used for all collected datasets is shown, with the Ca$^{2+}$-ADP-BeF$_3^-$ F-actin dataset as example. All maps are shown in the same orientation. The white scale bar shown on the micrograph is 400 Å. The box size of the 2D-class averages is 384x384 pixels (267x267 Å).

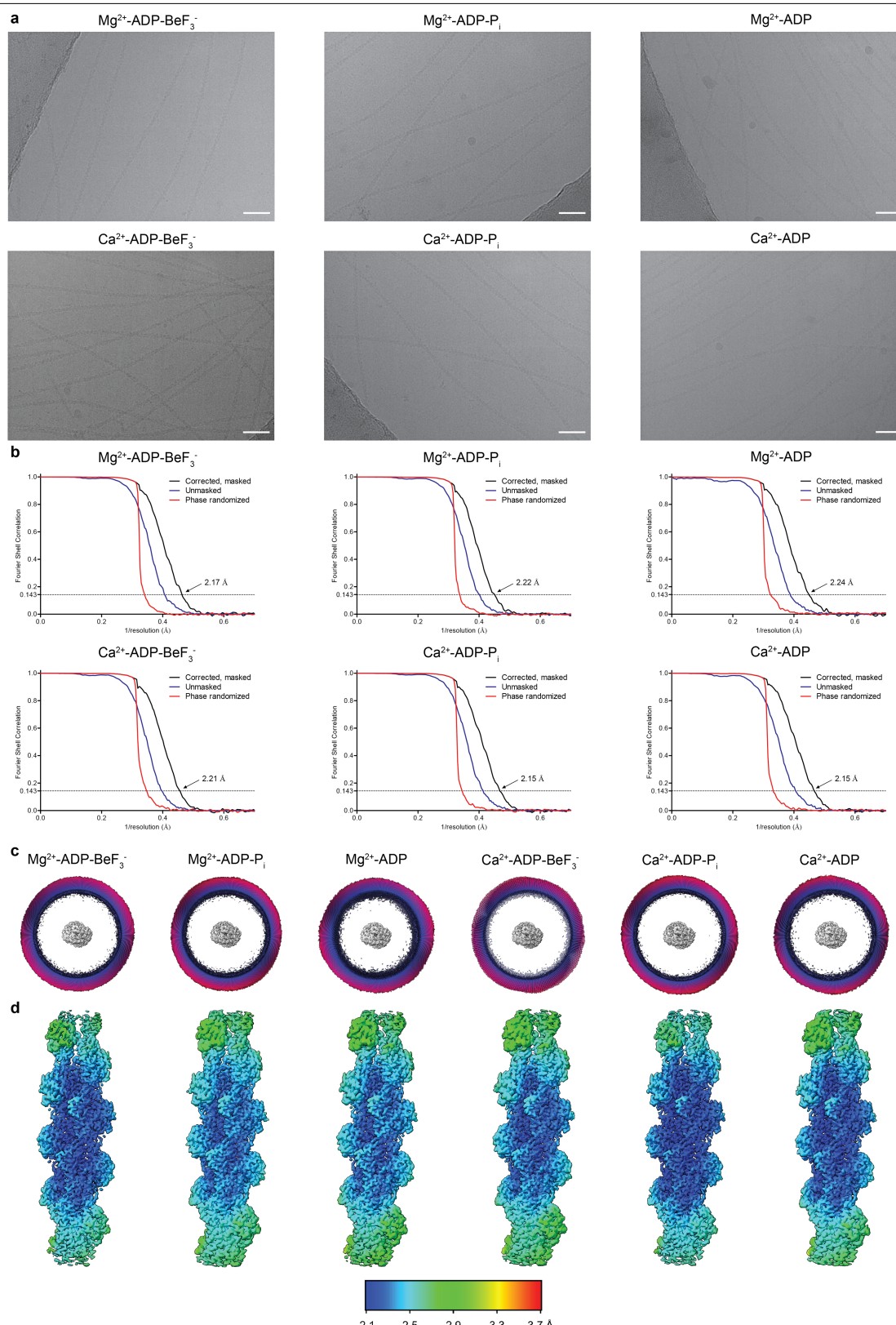

**Extended Data Fig. 2 | Image processing of all cryo-EM datasets.**
**a** Micrograph depicting actin filaments in the Mg$^{2+}$-ADP-BeF$_3^-$ (−2.1 μm),
Mg$^{2+}$-ADP-P$_i$ (−2.4 μm), Mg$^{2+}$-ADP (−1.3 μm), Ca$^{2+}$-ADP-BeF$_3^-$ (−1.4 μm), Ca$^{2+}$-ADP-P$_i$ (−2.0 μm) and Ca$^{2+}$-ADP (−1.0 μm) states distributed in vitreous ice (defocus values between brackets). The shown micrographs are example images from full datasets consisting of the following number of analysed micrographs (dataset between brackets): 10822 (Mg$^{2+}$-ADP-BeF$_3^-$), 9658 (Mg$^{2+}$-ADP-P$_i$), 9842 (Mg$^{2+}$-ADP), 10705 (Ca$^{2+}$-ADP-BeF$_3^-$), 10156 (Ca$^{2+}$-ADP-P$_i$), 10733 (Ca$^{2+}$-ADP). The white scale bar shown on each micrograph is 400 Å. **b** Fourier-shell correlation plots for each F-actin structure of gold-standard refined masked (black), unmasked (blue) and high-resolution phase randomized (red) half-maps. The FSC = 0.143 threshold is depicted as a dashed line. **c** Angular distribution of the particles used in the final reconstruction, shown along the filament axis. **d** Local-resolution estimations of the F-actin reconstructions, computed through Relion.

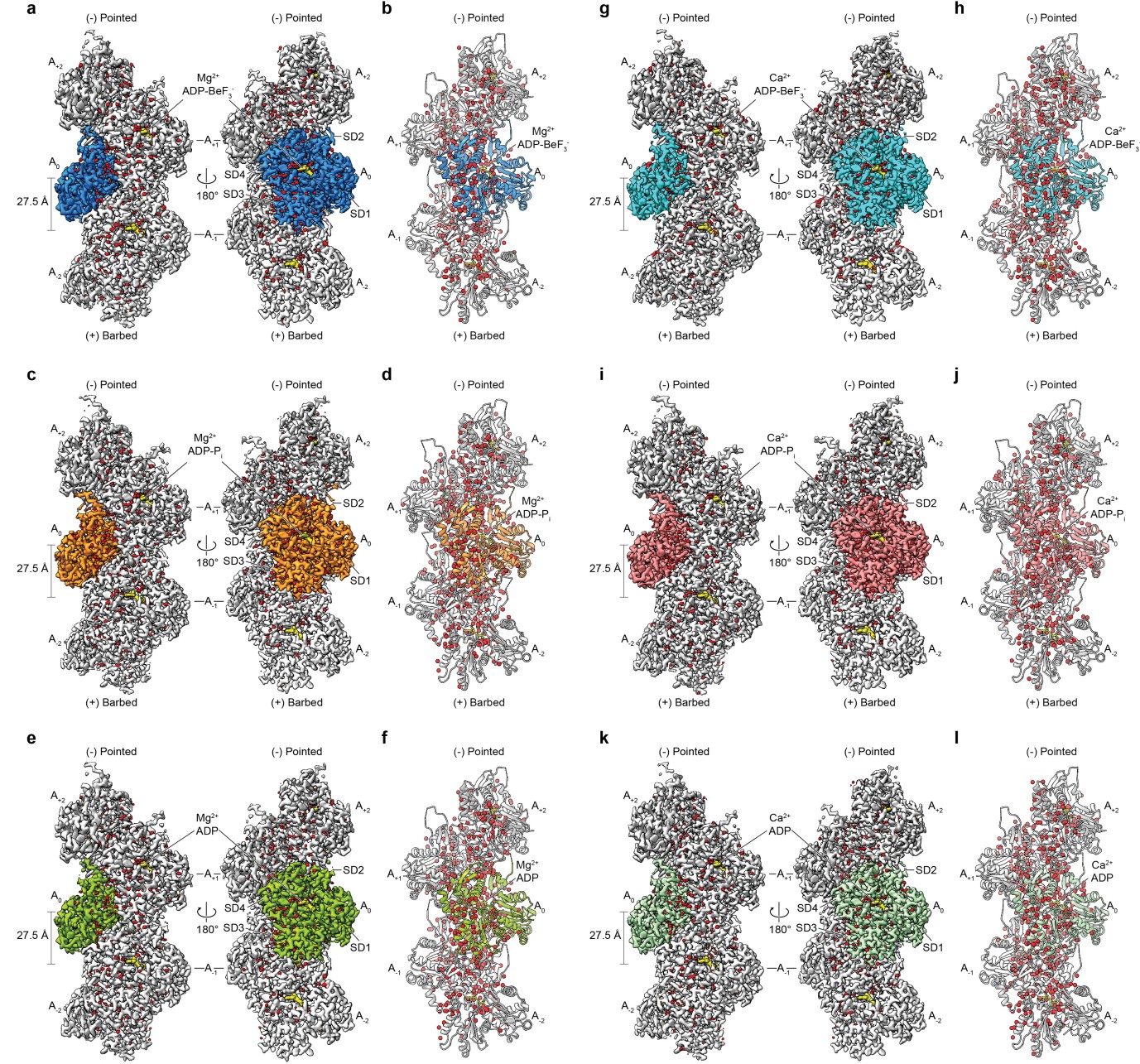

**Extended Data Fig. 3 | High-resolution cryo-EM structures of F-actin allow for the modelling of water molecules. a**, **c**, **e**, **g**, **i**, **k** Local-resolution filtered, sharpened cryo-EM density map of $Mg^{2+}$-ADP-$BeF_3^-$ (**a**), $Mg^{2+}$-ADP-$P_i$ (**c**), $Mg^{2+}$-ADP-F-actin (**e**), $Ca^{2+}$-ADP-$BeF_3^-$ (**g**), $Ca^{2+}$-ADP-$P_i$ (**i**) and $Ca^{2+}$-ADP-F-actin (**k**) shown in two orientations. The subunits are labelled based on their location along the filament, ranging from the barbed ($A_{-2}$) to the pointed ($A_2$) end. The central actin subunit ($A_0$) is coloured blue (**a**), orange (**c**), green (**e**), cyan (**g**), salmon (**i**) or pale green (**k**) the other four subunits are grey. Densities corresponding to water molecules are coloured red. **b**, **d**, **f**, **h**, **j**, **l** Cartoon representation of the of $Mg^{2+}$-ADP-$BeF_3^-$ (**b**), $Mg^{2+}$-ADP-$P_i$ (**d**), $Mg^{2+}$-ADP-F-actin (**f**), $Ca^{2+}$-ADP-$BeF_3^-$ (**h**), $Ca^{2+}$-ADP-$P_i$ (**j**) and $Ca^{2+}$-ADP-F-actin (**l**) structures. Waters are shown as spheres to emphasize their location.

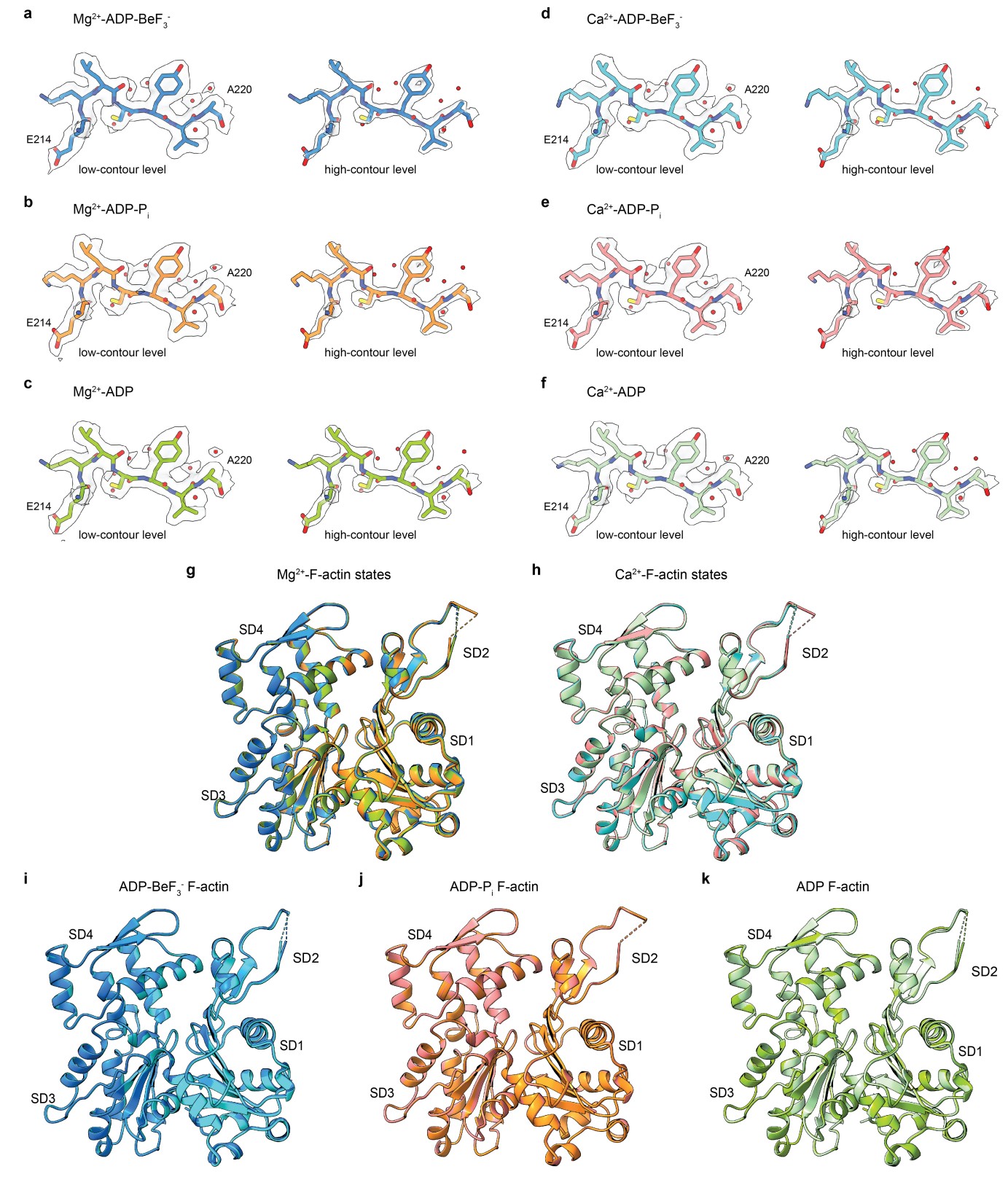

**Extended Data Fig. 4 | Modelling of selected regions and structural similarities between high-resolution F-actin structures. a–f** Cryo-EM density of residues E214 – A220 with modelled amino acids and water molecules of F-actin in the Mg$^{2+}$-ADP-BeF$_3^-$ (**a**), Mg$^{2+}$-ADP-P$_i$ (**b**), Mg$^{2+}$-ADP (**c**) Ca$^{2+}$-ADP-BeF$_3^-$ (**d**), Ca$^{2+}$-ADP-P$_i$ (**e**), Ca$^{2+}$-ADP (**f**) states, shown at two different contour levels. **g** Superimposition of a single subunit of Mg$^{2+}$-F-actin in ADP-BeF$_3^-$ (blue), ADP-P$_i$ (orange) and ADP (green) states. **h** Superimposition of a single subunit of Ca$^{2+}$-F-actin in ADP-BeF$_3^-$ (cyan), ADP-P$_i$ (salmon) and ADP (pale green) states. **i–k** Superimpositions of Mg$^{2+}$-F-actin and Ca$^{2+}$-F-actin in the ADP-BeF$_3^-$ (**i**), ADP-P$_i$ (**j**) and ADP (**k**) states. The colouring is consistent with the descriptions of **g** and **h**. For each overlay, the subdomains of actin (SD1 – SD4) are annotated.

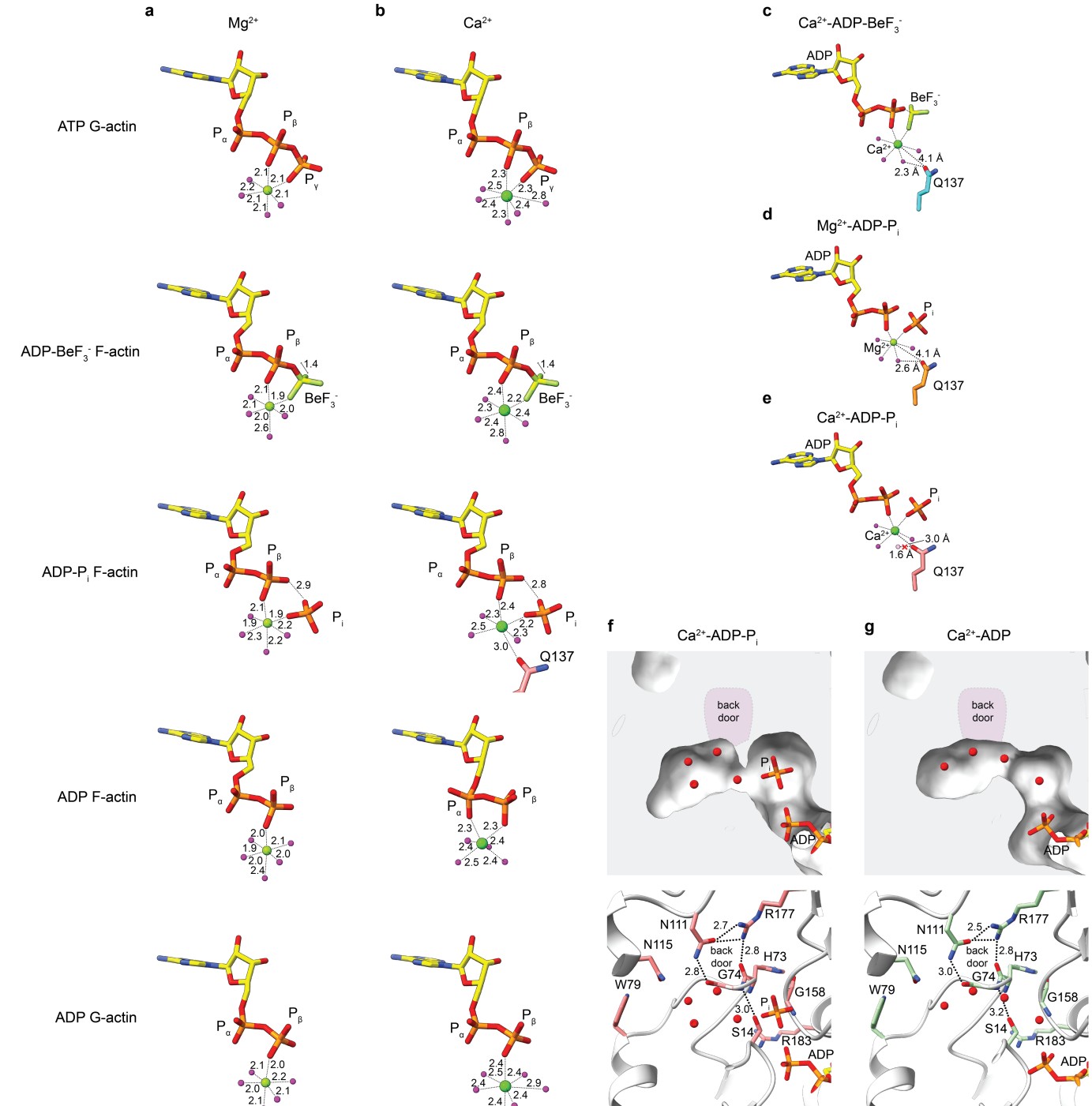

**Extended Data Fig. 5 | Ion coordination at the nucleotide-binding sites and $P_i$ release from $Ca^{2+}$-F-actin. a, b** Nucleotide conformation and inner-coordination sphere of the cation for $Mg^{2+}$- (**a**) and $Ca^{2+}$-actin (**b**). The shown G-actin models were selected from high-resolution crystal structures of rabbit G-actin in the following states: $Mg^{2+}$-ATP (pdb 2v52, 1.45 Å), $Mg^{2+}$-ADP (pdb 6rsw, 1.95 Å), $Ca^{2+}$-ATP (pdb 1qz5, 1.45 Å) and $Ca^{2+}$-ADP (pdb 1j6z, 1.54 Å). All distances are shown in Å. The distances between the cation and the molecules in its coordination sphere were not restrained during model refinement and may therefore deviate from ideal values. In the ADP-$BeF_3^-$-bound structures, the distance between the oxygen of the β-phosphate ($P_β$) of ADP and Be (1.4 Å) is as short as the equivalent distance in ATP (1.5 Å), defining ADP-$BeF_3^-$ as a mimic of the ATP ground state of F-actin, rather than an ADP-$P_i$-like state. **c**–**e** Position of the nucleotide, cation and associated waters with respect to residue Q137 in the

$Ca^{2+}$-ADP-$BeF_3^-$ (**c**), $Mg^{2+}$-ADP-$P_i$ (**d**) and $Ca^{2+}$-ADP-$P_i$ (**e**) states of F-actin. In the $Ca^{2+}$-ADP-$P_i$ state (panel **e**), the position of Q137 prevents the binding of one of the $Ca^{2+}$-coordinating waters (shown in semitransparent magenta), yielding an octahedral inner-coordination sphere of $Ca^{2+}$ with one missing water, but instead a coordination by Q137. **f, g** Internal solvent cavities near the $P_i$ binding site in ADP-$P_i$ (**f**) and ADP (**g**) structures of $Ca^{2+}$-F-actin. The upper panel shows the F-actin structure as surface with the bound $P_i$ and water molecules. In the lower panel, F-actin is shown in cartoon representation and the amino acids forming the internal cavity are annotated and shown as sticks. Hydrogen bonds are depicted as dashed line. All distances are shown in Å. The position of the proposed back door is highlighted in purple in the upper panel. In none of the structures, the internal solvent cavity is connected to the exterior milieu.

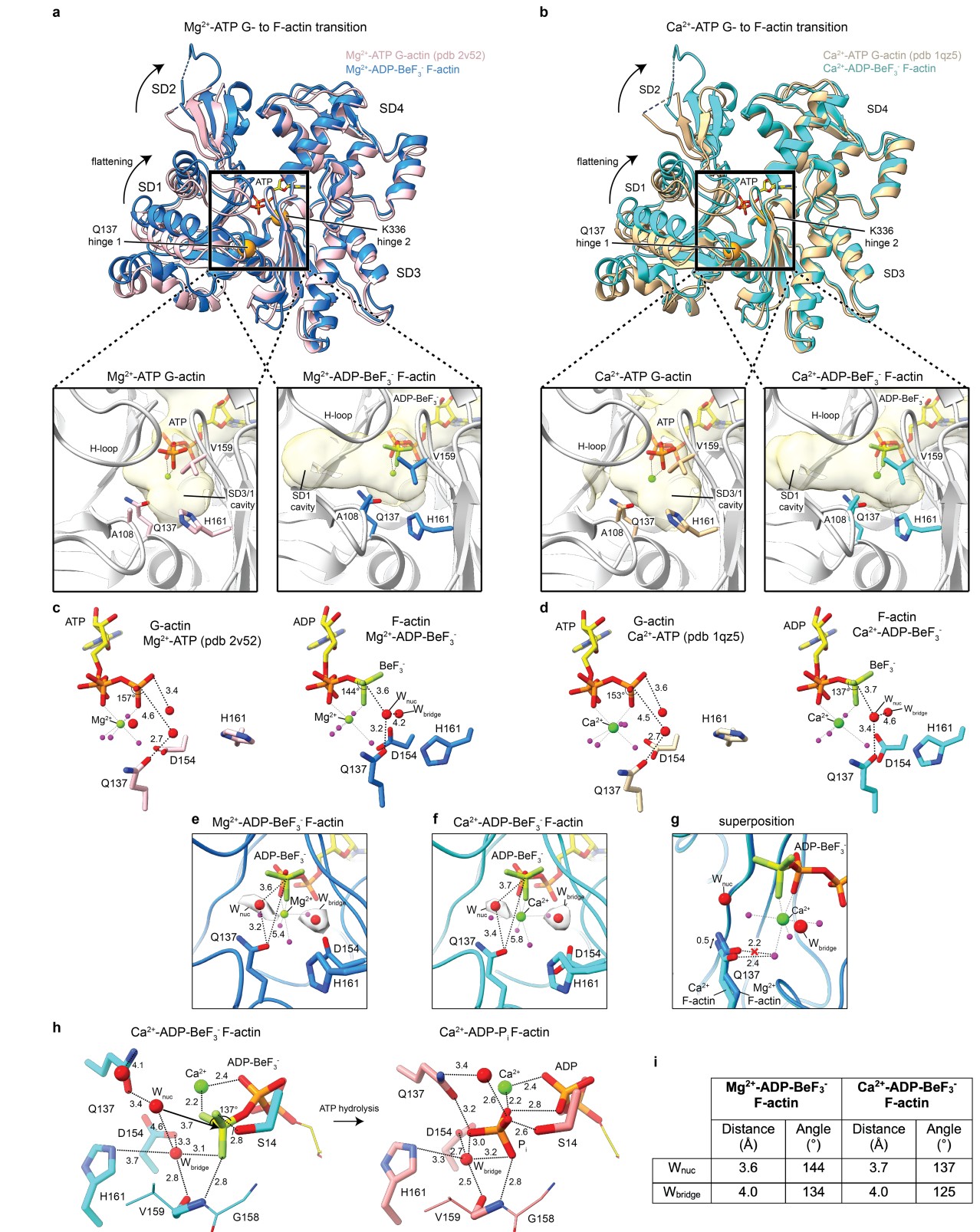

**Extended Data Fig. 6** | See next page for caption.

**Extended Data Fig. 6 | Rearrangements upon actin flattening and ATP hydrolysis. a**, **b** Upper panels: Overlay of G- and F-actin structures show the global conformational changes associated with actin flattening for $Mg^{2+}$- (**a**) and $Ca^{2+}$-F-actin (**b**). Residues Q137 and K336 act as hinges (as calculated by DynDom[67]) and are shown as orange spheres. Lower panels: Internal solvent cavities near the nucleotide-binding site in G- and F-actin. The cavities were calculated by the Castp3 server[64] and are shown as beige, semitransparent surfaces. The nucleotide was not considered in the solvent cavity calculations. **c**, **d** Water arrangement in front of the nucleotide in structures of $Mg^{2+}$-ATP-G-actin (pdb 2v52, left) and $Mg^{2+}$-ADP-$BeF_3^-$-F-actin (right) (**c**), and $Ca^{2+}$-ATP-G-actin (pdb 1qz5, left) and $Ca^{2+}$-ADP-$BeF_3^-$-F-actin (right) (**d**). The waters that coordinate the nucleotide-associated cation are coloured magenta, whereas the waters important for the hydrolysis mechanism are shown as larger red spheres. **e**, **f** front view of the $P\gamma$-mimic $BeF_3^-$ with densities for the putative $W_{nuc}$ and $W_{bridge}$ in structures of $Mg^{2+}$- (**e**) and $Ca^{2+}$-F-actin (**f**). **g** Overlay of the amino-acid arrangement in front of ADP-$BeF_3^-$ in $Mg^{2+}$- (blue) and $Ca^{2+}$-F-actin (cyan). The nucleotide arrangement of $Ca^{2+}$-F-actin is shown to emphasize that Q137 in its $Mg^{2+}$-F-actin conformation would clash with a water in the inner-coordination sphere of the $Ca^{2+}$-ion. (**h**) Mechanism of ATP hydrolysis in $Ca^{2+}$-F-actin. The isolated amino acid and water arrangement near the nucleotide in $Ca^{2+}$-ADP-$BeF_3^-$-F-actin (left) and $Ca^{2+}$-ADP-$P_i$-F-actin (right) are depicted. Regions unimportant for interactions are depicted as smaller sticks. Amino acids and $W_{nuc}$ and assisting water $W_{bridge}$ are annotated. (**i**) Table depicting the distance and angles of waters in the nucleotide-binding pocket to the Be-atom in structures of $Mg^{2+}$-ADP-$BeF_3^-$ and Ca-ADP-$BeF_3^-$-F-actin. The distances/angles were measured in the central subunit (chain c) of the reconstruction, which displays the highest local resolution. All annotated distances are shown in Å.

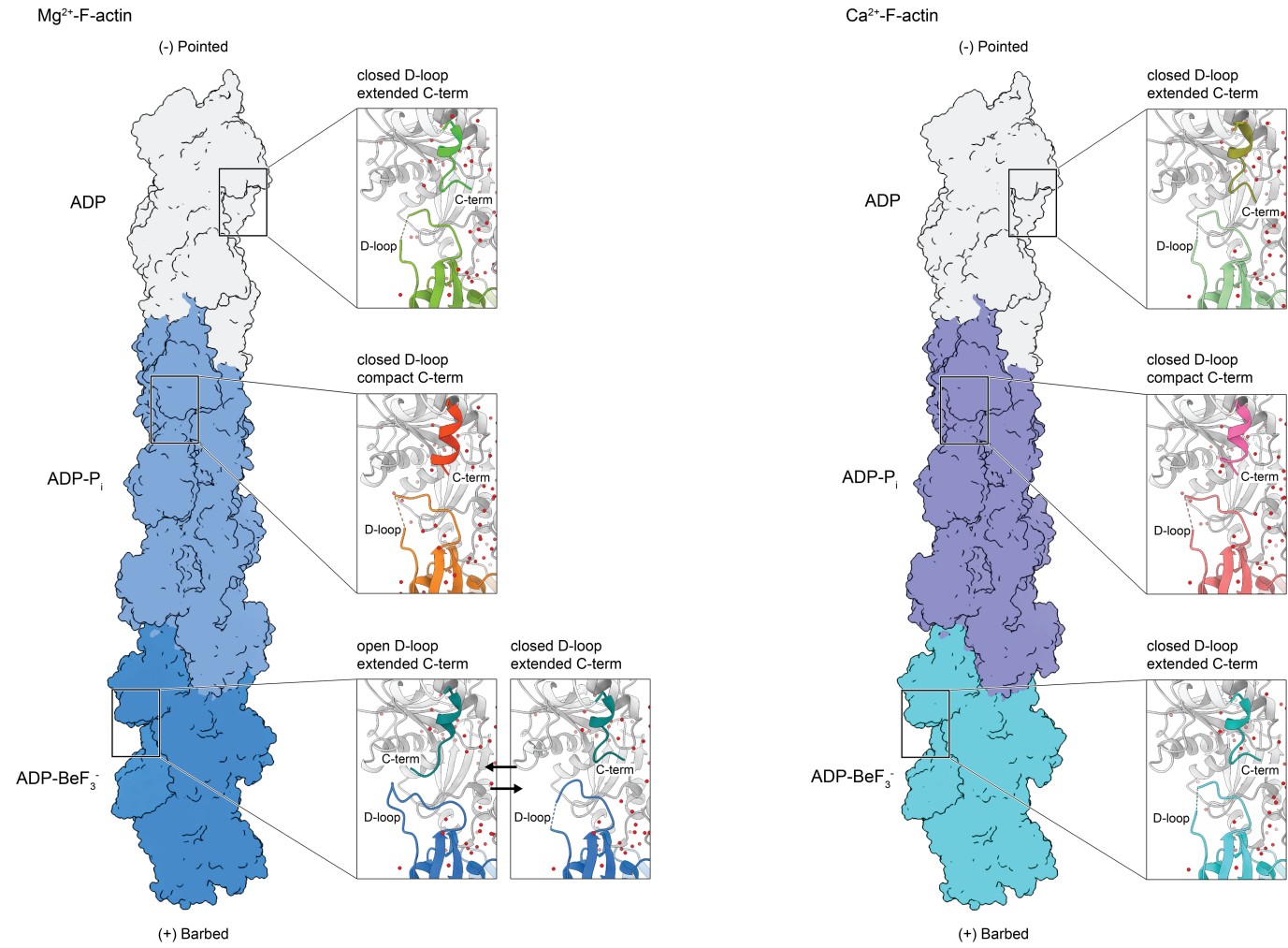

**Extended Data Fig. 7 | Arrangement of the F-actin intrastrand interface.**
Merged models of Mg²⁺-F-actin and Ca²⁺-F-actin in all nucleotide states.
The structures are shown as surface and should be regarded as an infinitely
long polymer. For each nucleotide state, a close-up of the observed intrastrand
interface is depicted. The open D-loop conformation is likely adopted to a small
extent in every F-actin nucleotide state. However, the D-loop is mostly closed
in all nucleotide states except the Mg²⁺-ADP-BeF₃⁻ state, where a mixed open/
closed conformation is observed.

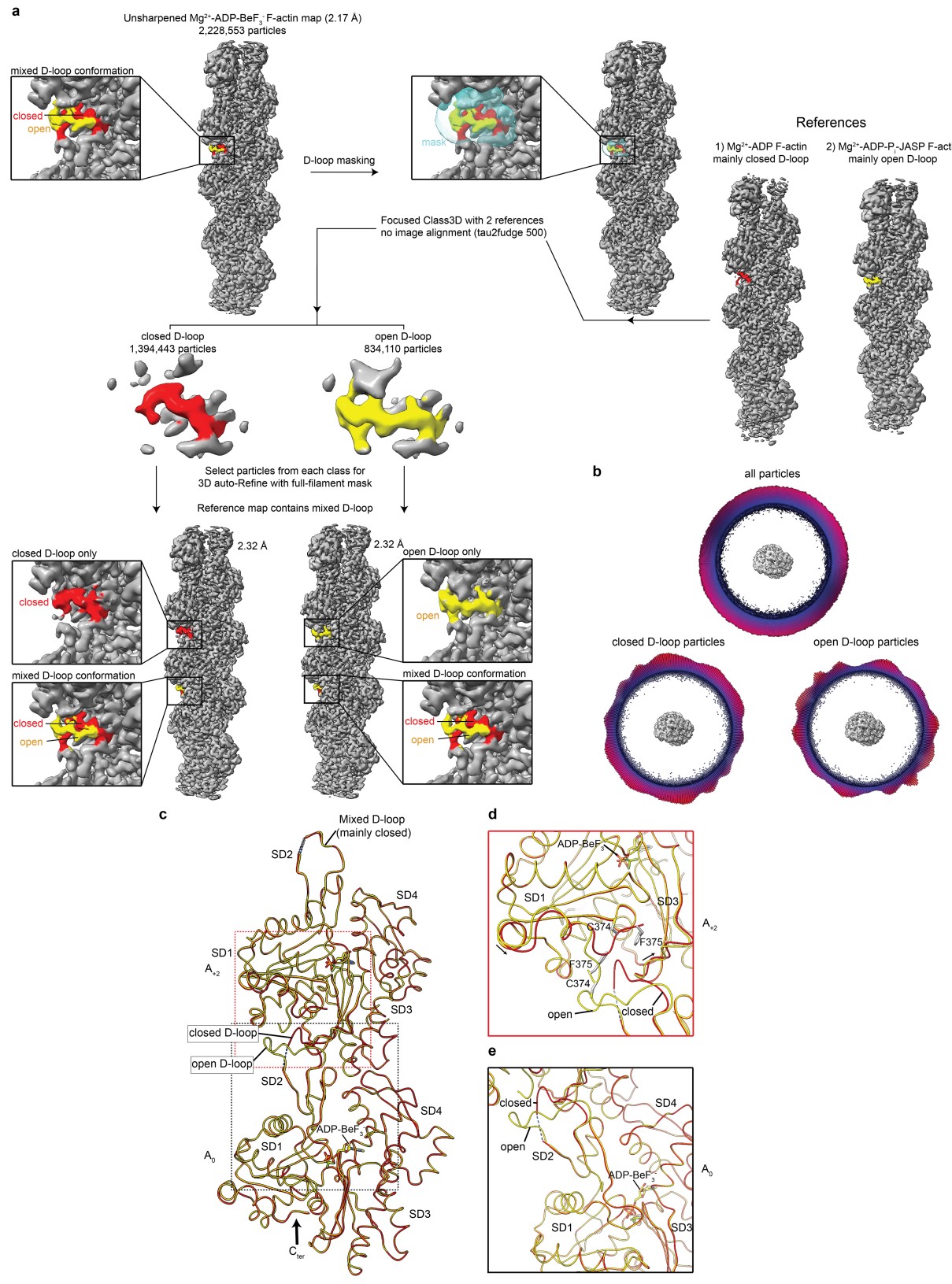

**Extended Data Fig. 8** | See next page for caption.

**Extended Data Fig. 8 | Conformational differences associated with open and closed D-loop. a** Focused classification strategy for the separation of the D-loop states of the $Mg^{2+}$-ADP-$BeF_3^-$ F-actin dataset. The closed D-loop is coloured red, whereas the open D-loop is coloured yellow. **b** Angular distribution of all particles used in the reconstruction of the $Mg^{2+}$-ADP-$BeF_3^-$ structure (top); and of the particles used for the reconstructions of the isolated closed (bottom-left) and open (bottom-right) D-loop. **c** Superposition of $Mg^{2+}$-ADP-$BeF_3^-$ F-actin structures with separated D-loop conformations in the central actin subunit ($A_0$). For both the open (yellow) and closed (red) D-loop states, the $A_0$ and $A_{+2}$ actin subunits are shown. **d, e** Close-up of the $A_0$ (**d**) and $A_{+2}$ (**e**) subunits. In panel **d**, arrows depict the movements in the SD1 and SD3 of the $A_{+2}$ subunit associated with the change of D-loop conformation from open to closed in the central $A_0$ subunit. The superposition of the $Mg^{2+}$-ADP-$BeF_3^-$ structures with the separated open and closed D-loop conformations in the central actin subunit, revealed that the only differences between the two structures, besides the D-loop, are not found in the central subunit, but instead in SD1 (including C terminus) and SD3 of the adjacent actin subunit. Our data therefore suggest that the D-loop conformation in an actin subunit is not affected by changes in the same subunit, but rather by changes in SD1 of its neighbour subunit.

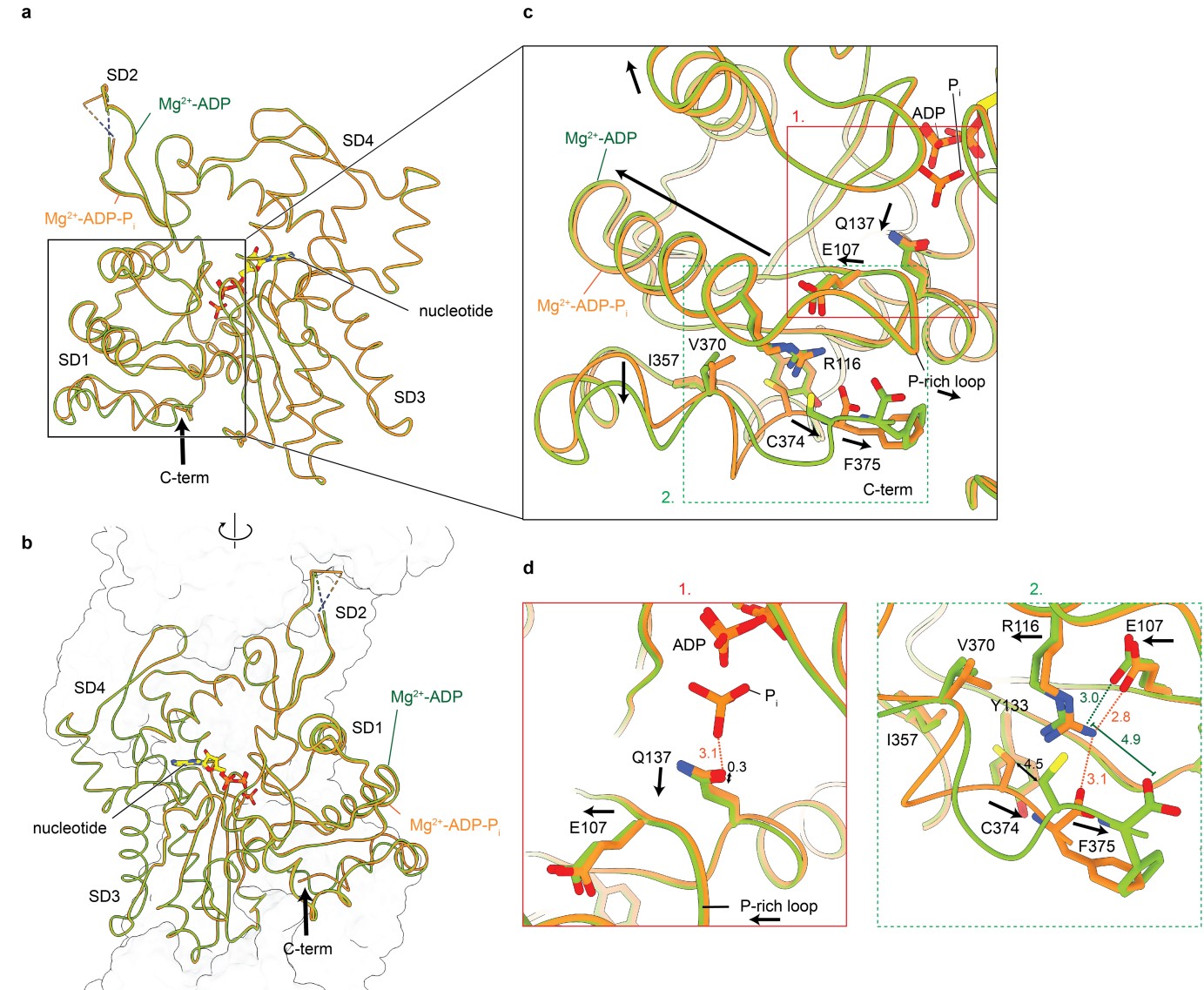

**Extended Data Fig. 9 | Structural coupling of the nucleotide-binding site to the filament exterior after P_i release. a**, **b** Overlay of one actin subunit in the $Mg^{2+}$-ADP-$P_i$ and $Mg^{2+}$-ADP structures with annotated subdomains, shown in two orientations. The location of the C terminus (C-term) is accentuated with a large arrow. In (**b**), the surface-contour of other actin subunits within the filament is depicted. **c** Differences in the SD1 of F-actin in the $Mg^{2+}$-ADP-$P_i$ and $Mg^{2+}$-ADP structures. Residues thought to be important for the movement are annotated. **d** Zoom of the nucleotide-binding site (1.) and C-terminal region (2.) of the SD1. In (**c**) and (**d**), arrows depict the direction of the putative movement from the $Mg^{2+}$-ADP-$P_i$ to the $Mg^{2+}$-ADP structure. All distances are shown in Å. Distances shown in the $Mg^{2+}$-ADP-$P_i$ structure are coloured orange, whereas those in the $Mg^{2+}$-ADP structure are coloured green.

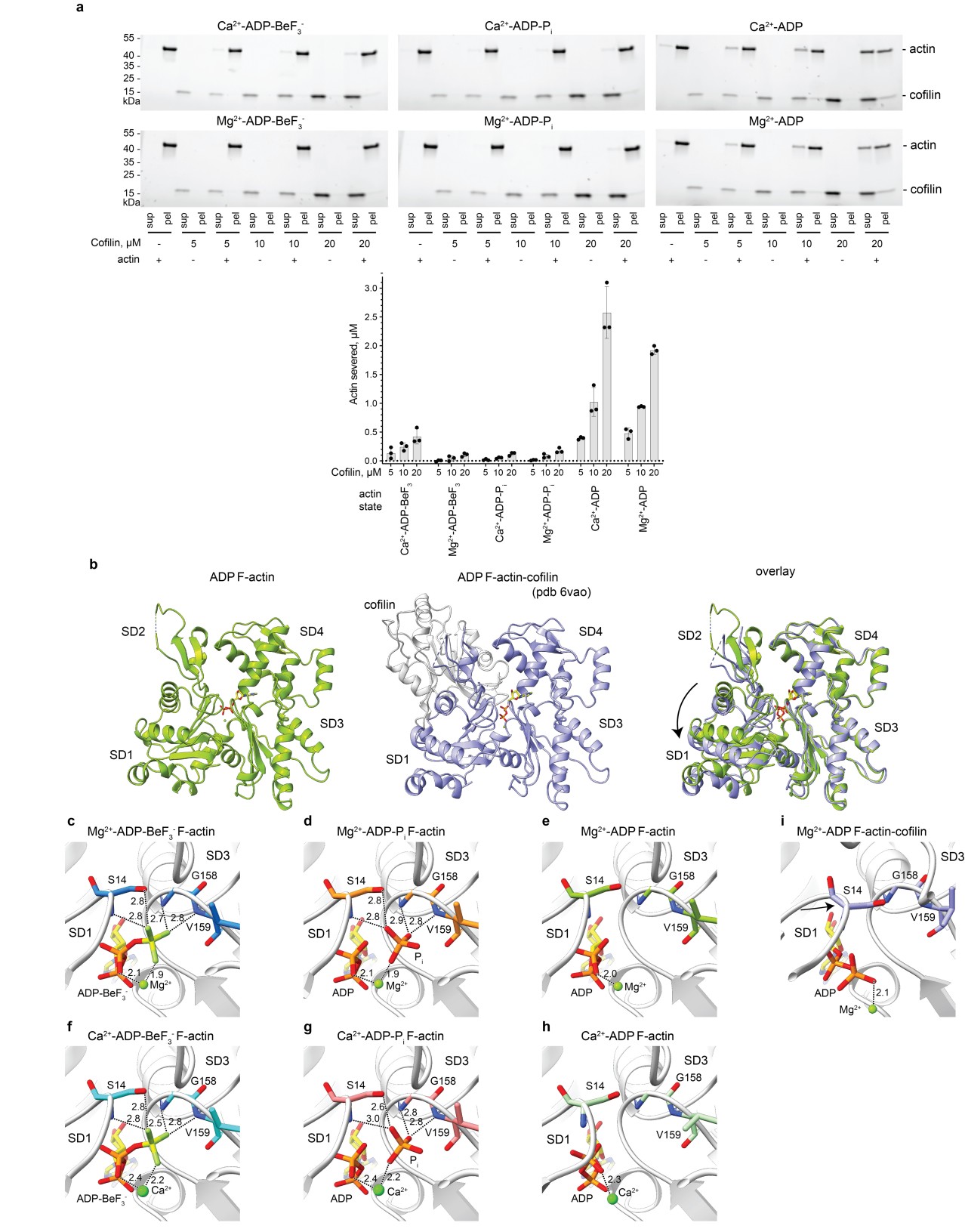

**Extended Data Fig. 10** | See next page for caption.

**Extended Data Fig. 10 | Cofilin-dependent F-actin severing and conformational changes. a**, top: Representative stain-free SDS–PAGE gel images of co-sedimentations of human cofilin-1 at 5, 10 or 20 μM concentrations with 5 μM $Ca^{2+}$-F-actin or $Mg^{2+}$-F-actin in ADP-$BeF_3^-$, ADP-$P_i$ and ADP states. The graph depicts the amount of F-actin severed by cofilin. Values were calculated from 3 independent assays. The proteins used in the assay originated from aliquots from the same batches of purified G-actin and cofilin-1. The data are presented as mean values. Error bars represent the standard deviation and were calculated in parallel from band intensities from the same experiment. Uncropped gel images are available in Supplementary Fig. 1. The data points used to obtain the graph are available as source data. Abbreviations: sup = supernatant, pel = pellet. **b** Structure of a single subunit of ADP-F-actin (left panel) and cofilin-decorated ADP-F-actin (middle panel). The right panel depicts an overlay between the two structures; cofilin is hidden for clarity. The arrow indicates the SD1 and SD2 rotation in F-actin upon cofilin binding. **c**–**i** Arrangement of the SD1 and SD3 at the nucleotide-binding sites in $Mg^{2+}$-ADP-$BeF_3^-$ (**c**), $Mg^{2+}$-ADP-$P_i$ (**d**), $Mg^{2+}$-ADP (**e**), $Ca^{2+}$-ADP-$BeF_3^-$ (**f**), $Ca^{2+}$-ADP-$P_i$ (**g**), $Ca^{2+}$-ADP (**h**), and cofilin-decorated $Mg^{2+}$-ADP (**i**) structures of F-actin. In panel **i**, the arrow depicts the cofilin−induced SD1 movement.

# Reporting Summary

## Statistics

For all statistical analyses, confirm that the following items are present in the figure legend, table legend, main text, or Methods section.

| n/a | Confirmed | |
|---|---|---|
| ☐ | ☒ | The exact sample size (*n*) for each experimental group/condition, given as a discrete number and unit of measurement |
| ☐ | ☒ | A statement on whether measurements were taken from distinct samples or whether the same sample was measured repeatedly |
| ☒ | ☐ | The statistical test(s) used AND whether they are one- or two-sided *Only common tests should be described solely by name; describe more complex techniques in the Methods section.* |
| ☒ | ☐ | A description of all covariates tested |
| ☒ | ☐ | A description of any assumptions or corrections, such as tests of normality and adjustment for multiple comparisons |
| ☐ | ☒ | A full description of the statistical parameters including central tendency (e.g. means) or other basic estimates (e.g. regression coefficient) AND variation (e.g. standard deviation) or associated estimates of uncertainty (e.g. confidence intervals) |
| ☒ | ☐ | For null hypothesis testing, the test statistic (e.g. *F*, *t*, *r*) with confidence intervals, effect sizes, degrees of freedom and *P* value noted *Give P values as exact values whenever suitable.* |
| ☒ | ☐ | For Bayesian analysis, information on the choice of priors and Markov chain Monte Carlo settings |
| ☒ | ☐ | For hierarchical and complex designs, identification of the appropriate level for tests and full reporting of outcomes |
| ☒ | ☐ | Estimates of effect sizes (e.g. Cohen's *d*, Pearson's *r*), indicating how they were calculated |

*Our web collection on statistics for biologists contains articles on many of the points above.*

## Software and code

Policy information about availability of computer code

| | |
|---|---|
| Data collection | Cryo-EM data was collected using the commercially available software EPU version 2.8 (Thermofisher Scientific). |
| Data analysis | Cryo-EM data collection was monitored and preprocessed on the fly using TranSPHIRE version 1.5.13. The preprocessing steps in TranSPHIRE involved gain and drift correction using UCSF MotionCor2 v1.3.0, CTF estimation with CTFFIND4 v4.1.13, and particle picking using SPHIRE-crYOLO v1.5.8. The data was further processed using helical SPHIRE v1.4 and RELION v3.1.0. Protein model building was performed in COOT v0.8.9.2 and the models were refined using phenix real-space refine v1.20.1-4487. Solvent cavities in protein structures were calculated through the CASTp 3.0 web server. Protein models were validated within the phenix suite v1.20.1-4487. Figures and videos that depict cryo-EM density maps and protein structures were prepared using UCSF ChimeraX v1.3.<br><br>Densitometry of SDS PAGE gel bands in cofilin-severing assays was performed with Image Lab software v6.0.1 (Bio-Rad) and the analysis was done in GrapPad Prism v9 (GraphPad Software). |

For manuscripts utilizing custom algorithms or software that are central to the research but not yet described in published literature, software must be made available to editors and reviewers. We strongly encourage code deposition in a community repository (e.g. GitHub). See the Nature Portfolio guidelines for submitting code & software for further information.

## Data

Policy information about availability of data

All manuscripts must include a data availability statement. This statement should provide the following information, where applicable:
- Accession codes, unique identifiers, or web links for publicly available datasets
- A description of any restrictions on data availability
- For clinical datasets or third party data, please ensure that the statement adheres to our policy

The cryo-EM maps have been deposited to the Electron Microscopy Data Bank (EMDB) under accession codes (dataset in brackets): EMD-15104 (Mg2+-ADP-BeF3-) [https://www.ebi.ac.uk/emdb/EMD-15104], EMD-15105 (Mg2+-ADP-Pi) [https://www.ebi.ac.uk/emdb/EMD-15105], EMD-15106 (Mg2+-ADP) [https://www.ebi.ac.uk/emdb/EMD-15106], EMD-15107 (Ca2+-ADP-BeF3-) [https://www.ebi.ac.uk/emdb/EMD-15107], EMD-15108 (Ca2+-ADP-Pi) [https://www.ebi.ac.uk/emdb/EMD-15108] and EMD-15109 (Ca2+-ADP) [https://www.ebi.ac.uk/emdb/EMD-15109]. These depositions include sharpened maps, unfiltered half-maps and the refinement masks. For the Mg2+-ADP-BeF3- F-actin submission, all density maps and masks regarding the separation of open/closed D-loop conformations are provided. The atomic coordinates of the protein structures have been submitted to the Protein Data Bank (PDB) under accession codes (dataset in brackets): 8A2R (Mg2+-ADP-BeF3-) [https://doi.org/10.2210/pdb8A2R/pdb], 8A2S (Mg2+-ADP-Pi) [https://doi.org/10.2210/pdb8A2S/pdb], 8A2T (Mg2+-ADP) [https://doi.org/10.2210/pdb8A2T/pdb], 8A2U (Ca2+-ADP-BeF3-) [https://doi.org/10.2210/pdb8A2U/pdb], 8A2Y (Ca2+-ADP-Pi) [https://doi.org/10.2210/pdb8A2Y/pdb] and 8A2Z (Ca2+-ADP) [https://doi.org/10.2210/pdb8A2Z/pdb]. We used the following previously published structures for modeling and comparisons: 7AHN [https://doi.org/10.2210/pdb7AHN/pdb], 2V52 [https://doi.org/10.2210/pdb2V52/pdb], 6RSW [https://doi.org/10.2210/pdb6RSW/pdb], 1QZ5 [https://doi.org/10.2210/pdb1QZ5/pdb], 1J6Z [https://doi.org/10.2210/pdb1J6Z/pdb]. EMD-11787 [https://www.ebi.ac.uk/emdb/EMD-11787] was used as initial model for the first 3D refinement. Source data are provided with this paper.

# Field-specific reporting

Please select the one below that is the best fit for your research. If you are not sure, read the appropriate sections before making your selection.

☒ Life sciences          ☐ Behavioural & social sciences          ☐ Ecological, evolutionary & environmental sciences

For a reference copy of the document with all sections, see nature.com/documents/nr-reporting-summary-flat.pdf

# Life sciences study design

All studies must disclose on these points even when the disclosure is negative.

| | |
|---|---|
| Sample size | Sample sizes for the six cryo-EM datasets presented in this study: For the Mg-ADP-BeF3- F-actin dataset, 10,822 micrographs were collected. 2,897,679 total particles were picked and 2,228,553 particles were used for the final reconstruction. For the Mg-ADP-Pi F-actin dataset, 9,658 micrographs were collected. 2,349,979 total particles were picked and 1,808,554 particles were used for the final reconstruction. For the Mg-ADP F-actin dataset, 9,842 micrographs were collected. 1,296,776 total particles were picked and 1,114,051 particles were used for the final reconstruction. For the Ca-ADP-BeF3- F-actin dataset, 10,705 micrographs were collected. 2,246,622 total particles were picked and 1,719,432 particles were used for the final reconstruction. For the Ca-ADP-Pi F-actin dataset, 10,156 micrographs were collected. 3,031,270 total particles were picked and 2,171,987 particles were used for the final reconstruction. For the Ca-ADP F-actin dataset, 10,733 micrographs were collected. 1,873,773 total particles were picked and 1,073,455 particles were used for the final reconstruction. These sample sizes of ~10,000 micrographs are common in the cryo-EM field for obtaining high-resolution protein structures, see for example Belyy et al. Nat. Commun. (2021): https://doi.org/10.1038/s41467-021-26889-2.<br><br>The cofilin-severing assays were performed as three independent experiments. The sample size of n=3 is common for in vitro assays with purified proteins, see for example Belyy et al. Plos Biol. (2020): https://doi.org/10.1371/journal.pbio.3000925 |
| Data exclusions | During the cryo-EM image processing, particles that represented false picks or particles that did not contribute high-resolution information to the reconstructions were discarded through 2D and 3D classification procedures. This process, which is required to obtain high-resolution reconstructions, is a standard procedure in cryo-EM image processing. |
| Replication | All cryo-EM datasets were collected in one session per F-actin functional state and were not repeated. It is unattainable from a time and cost perspective to repeat cryo-EM data collection and processing on the exact same sample.<br><br>The cofilin-severing assays were performed in triplicate. They were performed as independent experiments, with new protein aliquots from the same purification batch.<br>All attempts at replication were successful. |
| Randomization | For the 3D refinement of cryo-EM structures, particles were randomly split into two half sets. For all other experiments, randomization was not required because all data were used in the analysis. Covariates were not controlled. |
| Blinding | This study does not involve any experiments where blinding would be applicable. |

# Behavioural & social sciences study design

All studies must disclose on these points even when the disclosure is negative.

| | |
|---|---|
| Study description | *Briefly describe the study type including whether data are quantitative, qualitative, or mixed-methods (e.g. qualitative cross-sectional, quantitative experimental, mixed-methods case study).* |
| Research sample | *State the research sample (e.g. Harvard university undergraduates, villagers in rural India) and provide relevant demographic information (e.g. age, sex) and indicate whether the sample is representative. Provide a rationale for the study sample chosen. For studies involving existing datasets, please describe the dataset and source.* |
| Sampling strategy | *Describe the sampling procedure (e.g. random, snowball, stratified, convenience). Describe the statistical methods that were used to predetermine sample size OR if no sample-size calculation was performed, describe how sample sizes were chosen and provide a rationale for why these sample sizes are sufficient. For qualitative data, please indicate whether data saturation was considered, and what criteria were used to decide that no further sampling was needed.* |
| Data collection | *Provide details about the data collection procedure, including the instruments or devices used to record the data (e.g. pen and paper, computer, eye tracker, video or audio equipment) whether anyone was present besides the participant(s) and the researcher, and whether the researcher was blind to experimental condition and/or the study hypothesis during data collection.* |
| Timing | *Indicate the start and stop dates of data collection. If there is a gap between collection periods, state the dates for each sample cohort.* |
| Data exclusions | *If no data were excluded from the analyses, state so OR if data were excluded, provide the exact number of exclusions and the rationale behind them, indicating whether exclusion criteria were pre-established.* |
| Non-participation | *State how many participants dropped out/declined participation and the reason(s) given OR provide response rate OR state that no participants dropped out/declined participation.* |
| Randomization | *If participants were not allocated into experimental groups, state so OR describe how participants were allocated to groups, and if allocation was not random, describe how covariates were controlled.* |

# Ecological, evolutionary & environmental sciences study design

All studies must disclose on these points even when the disclosure is negative.

| | |
|---|---|
| Study description | *Briefly describe the study. For quantitative data include treatment factors and interactions, design structure (e.g. factorial, nested, hierarchical), nature and number of experimental units and replicates.* |
| Research sample | *Describe the research sample (e.g. a group of tagged Passer domesticus, all Stenocereus thurberi within Organ Pipe Cactus National Monument), and provide a rationale for the sample choice. When relevant, describe the organism taxa, source, sex, age range and any manipulations. State what population the sample is meant to represent when applicable. For studies involving existing datasets, describe the data and its source.* |
| Sampling strategy | *Note the sampling procedure. Describe the statistical methods that were used to predetermine sample size OR if no sample-size calculation was performed, describe how sample sizes were chosen and provide a rationale for why these sample sizes are sufficient.* |
| Data collection | *Describe the data collection procedure, including who recorded the data and how.* |
| Timing and spatial scale | *Indicate the start and stop dates of data collection, noting the frequency and periodicity of sampling and providing a rationale for these choices. If there is a gap between collection periods, state the dates for each sample cohort. Specify the spatial scale from which the data are taken* |
| Data exclusions | *If no data were excluded from the analyses, state so OR if data were excluded, describe the exclusions and the rationale behind them, indicating whether exclusion criteria were pre-established.* |
| Reproducibility | *Describe the measures taken to verify the reproducibility of experimental findings. For each experiment, note whether any attempts to repeat the experiment failed OR state that all attempts to repeat the experiment were successful.* |
| Randomization | *Describe how samples/organisms/participants were allocated into groups. If allocation was not random, describe how covariates were controlled. If this is not relevant to your study, explain why.* |
| Blinding | *Describe the extent of blinding used during data acquisition and analysis. If blinding was not possible, describe why OR explain why blinding was not relevant to your study.* |

Did the study involve field work? 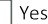 Yes 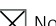 No

# Reporting for specific materials, systems and methods

We require information from authors about some types of materials, experimental systems and methods used in many studies. Here, indicate whether each material, system or method listed is relevant to your study. If you are not sure if a list item applies to your research, read the appropriate section before selecting a response.

## Materials & experimental systems

| n/a | Involved in the study |
|---|---|
| ☒ | ☐ Antibodies |
| ☒ | ☐ Eukaryotic cell lines |
| ☒ | ☐ Palaeontology and archaeology |
| ☒ | ☐ Animals and other organisms |
| ☒ | ☐ Human research participants |
| ☒ | ☐ Clinical data |
| ☒ | ☐ Dual use research of concern |

## Methods

| n/a | Involved in the study |
|---|---|
| ☒ | ☐ ChIP-seq |
| ☒ | ☐ Flow cytometry |
| ☒ | ☐ MRI-based neuroimaging |

