## [Peer Review File · Nature]

Manuscript Title: Structural basis of actin filament assembly and aging

Reviewer Comments & Author Rebuttals

Reviewer Reports on the Initial Version:

Referees' comments:

Referee #1 (Remarks to the Author):

Actin cytoskeleton dynamics are essential in many cellular processes, including motility, differentiation and division, and are perturbed in a number of disease contexts. The transitions between monomeric (G-actin) and filamentous (F-actin) states are regulated by bound nucleotide and associated cations, together with a large number of actin-binding proteins. Within F-actin, ATP hydrolysis followed by release of inorganic phosphate (Pi) are known to induce small conformational changes that in turn cause instability and ultimately depolymerization of F-actin.

Our understanding of the structural biology and biochemistry of actin are supported by decades of research. The current manuscript provides new data that build on existing understanding and provide explanations for several long-standing observations in the field. Using high resolution cryo-EM and ATP analogues, the authors investigate the structural consequences of ATP hydrolysis and phosphate release in rabbit skeletal F-actin in the presence of Ca²⁺ or Mg²⁺. Overall, the structures determined in these different states are extremely similar. However, the resolution achieved (~2.2-Å) allows precise description small rearrangements including of ordered water molecules within the actin/nucleotide complex. From these insights, the authors infer a role for specific water molecules in the ATP hydrolysis mechanism and also explain slower polymerization rates of F-actin in the presence of Ca²⁺. From the similarity of the structures in the ADP.Pi and ADP nucleotide binding sites, the authors infer that a transient – and thus not captured – conformational change in an actin “back door” is involved in Pi release. They also describe a number of longer-range inter-subunit nucleotide-dependent conformational variations between Mg²⁺ F-actin and Ca²⁺ F-actin. It was previously assumed that the well-characterised recognition of ADP-F-actin by the cytoskeleton regulator cofilin was mediated through nucleotide-dependent conformational changes at its binding site at high radius on F-actin. There is no evidence for such changes in the authors' data – rather, they now suggest the intriguing alternative hypothesis that cofilin recognition/binding is mediated through intrinsically greater flexibility in ADP-F-actin due to loss of structural connectivity mediated by g-phosphate between actin SD1 and SD3.

In general, the data are clearly presented with a good balance of information in the main figures and extended data, and the manuscript is very well written and organised. Apart from point 4 below, there is appropriate use of statistics and treatment of uncertainties and apart from point 5 below, the conclusions appear reliable. Cytoskeleton enthusiasts will be satisfied by the many new details that emerge from these structures. The following points would expand the impact of the work and improve the methodological clarity of the study:

1. The authors use well-established nucleotide ATP-like analogues to infer the ATP F-actin structural state. However, at the resolutions at which the authors are working and at the level of individual water molecules, can they really be certain that these analogues are truly structurally analogous at atomic resolution? It would be very exciting and novel to see the authors apply their processing methods to dynamic actin filaments, enabling them to capture physiologically relevant nucleotide-dependent structural transitions that do not rely on analogues.

2. The authors state (p10) that their data show that the Ca^{2+} ion is more mobile within the actin catalytic site and that this could decrease filament stability and explain these filaments' higher depolymerization rate. It is not clear how reduced stability arises from ion mobility. Does this reduced stability manifest in their structures? If not, why not?

3. As depicted in Figure 5, evidence for the actin subunit back door by which P_i is released is inferred to be formed by the side chains of R177 and N111 and the backbones of methylated histidine 73 (H73) and G74, and is in a closed conformation in both ADPPi and ADP states of Mg^{2+} -F-actin and Ca^{2+} of F-actin. Can more extensive EM data processing in this area provide more direct evidence for conformational change in this region? Could the authors include molecular dynamics experiments to provide evidence about the behaviour of the back door?

4. In figure 6, several rearrangements of the D-loop and the C-terminus at the intrastrand interface in the actin filament, in the presence of Ca^{2+} and Mg^{2+} , are described, but more information is needed about the methodology used during the focused classification steps. Specifically, the authors should clearly explain why and how they used two initial model in order to classify the particles. In particular, how did they mitigate the possibility of model bias arising from the use of high value of T (500) combined with a high-resolution filtering of initial models (4 Å). Could alternative image processing strategies – e.g. with the use of signal subtraction combined with mask classification and refinement – provide additional validation of the structural interpretations arising from these analyses?

5. The results of the cofilin cosedimentation assay in the presence of Ca^{2+} -F-actin are used to derive a new nucleotide-dependent recognition mechanism for cofilin binding. The relevant text (p12-13) only refers to Ca^{2+} data but Mg^{2+} data are also presented in ED16/17 figure and should be incorporated into the description. Only a single concentration of cofilin is used in these experiments – do the observations hold up when a more complete titration is performed? This is important given the cooperative nature of cofilin binding to F-actin.

Referee #2 (Remarks to the Author):

Review: Oosterheert et al., "Structural basis of actin filament assembly and aging", Nature 422981_1, May 2022

The authors have determined filamentous F-actin structures in six different states (ADP-BeF₃, ADP-Pi and ADP, Mg- and Ca-bound, each) at true, near-atomic resolutions, close to 2 Å. Together with

previous high-resolution crystal structures of monomeric G-actin, this has enabled the authors to describe conformational changes and water molecule positions, and to make suggestions relating to the mechanisms underlying ATP hydrolysis, the G- to F-actin transition, the calcium effect, Pi release and the coupling of polymerisation state to actin-binding proteins.

While no entirely new methodology was invented or used, the combination of state-of-the-art sample preparation and cryo-EM data acquisition and data processing yielded maps of truly unprecedented quality of one of the most important molecule in the whole of eukaryotic biology, F-actin. The data are impressive, convincing and important. The level of detail obtained justifies the great majority of the claims made, and provides atomic snapshots that will make it possible to determine the molecular mechanisms of actin with absolute certainty, based on the structures and suggestions provided by the authors.

The paper is well written but will need to be re-formatted for Nature. I would suggest presenting hypothetical mechanisms in more cautious language throughout. Also, the number of main figures needs to be reduced, since too much detail is presented in my opinion. The authors might want to remove some of the data on calcium-bound actin from the main figures, for example to increase brevity and clarity.

I think this is definite landmark work in the field of the cytoskeleton (where only filaments that crystallise have reached this level of detail), the work presents a technical advance for cryo-EM, and is of significant interest to those working on, and thinking about atomic mechanisms of enzymes, which is very many people. It is also a showcase for cryo-EM not being a low-resolution method anymore, and that this is now true for many, if not most of samples. The work will also lead to many structures, at high resolutions, of important molecule bound to actin filaments.

A few specific points in no particular order:

- p1: Title: "aging" a good word, seems actin specific? "ATP hydrolysis" instead?
- intro: Mention the other subdomain nomenclature at least once (Ia, Ib, IIa, IIb)?
- general: Is looking at calcium-bound actin interesting? Given that not normally bound in cells (intro, p3). It is of course interesting in terms of understanding the mechanism better, but it might need to be phrased as such.
- general: Intra-strand contact is sometimes also called longitudinal.
- general: Did you consider/try using a transition-state mimic, such as aluminium fluoride (need to admit that I am not sure this works on actin)?
- p4: "All three functional states": is it not possible that there are more functional states that we can't trap? For example, the state where Pi leaves the active site? (see below, p10, missing open "back door" state).

- p6, top: I do not find it “surprising” that there are no extra binding sites for Mg and Pi.
- p6: How can we be sure that F-actin is bound to calcium and not a mix of magnesium and calcium (also: methods)?
- p8: What are the attack angles of the other two water and their distances (list)?
- p8 I find the attack angle of Wnuc slightly worrying, given the speed of the reaction involved. Could this have to do with the use of BeF₃, which will cause slightly altered geometry and dimensions? Any way to get to true ATP actin?
- p8: The putative mechanism of water attack and proton transfer is hypothetical at this point I would suggest. Please phrase as such. This applies to many hypothetical scenarios predicted from the structures, more cautious wording is warranted, generally.
- p8: Q137 and Q137/D154/H161 could have been investigated in more details with mutants.
- p9: The putative explanation of slower polymerisation and hydrolysis by calcium actin is quite convincing.
- p10: It is somewhat puzzling that the Pi “back door” is not visible in the ADP-Pi or ADP F-actin structures. See below, how the ADP-Pi state was generated.
- p11 & Movie S4: very minor changes are being discussed ... how can we be confident these are real differences, especially given that all atomic model building progressed from one model as far as I understood?
- p12: Does rabbit actin have all the modifications as human actin (since human cofilin was used)?
- p13: Not everything that affects rates is visible in atomic structures. For example, electrostatic interactions can drive affinities very significantly (Ca vs Mg / ATP vs ADP actin). ADP F-actin can have a much lower longitudinal affinity, purely based on changed electrostatics (2 vs 3 phosphates). This is thought to drive de-polymerisation in tubulin-like proteins, for example.
- p15: I note in the methods that the ADP-Pi state was created by adding Pi to ADP-actin, not by hydrolysis. That state could be different from a state that resulted from hydrolysis of ATP, and this may have shown the Pi “back door” open? I am aware that hydrolysis is much more difficult to control.
- p17: What particle, CTF and aberration parameters were refined in Bayesian polishing in RELION?
- p18: For the water molecules: was chemical plausibility taken into account when deciding on the positions of water molecules? How were waters distinguished from ions bound to the proteins? Some waters are not very round ... Crystallography has developed many tools for this to be done more objectively ...

- Figure 1: Very well presented. I would have added stereo versions of these as supplement.
- Figure 2: No bond is shown here for Be to Pbeta?
- Figure 4a: Again, what is the distance of Wbridge to Be? To me, the attack angle looks better for Wbridge? (The atomic model and/or stereo would help the reader to form an opinion).
- I think there are too many figures ...
- As far as I can see, only preliminary PDB validation reports have been provided. I thought submissions required full validation reports and PDB IDs these days to make sure structures have actually been submitted and accepted, before manuscript review?
- Line numbers would have been helpful throughout.
- A personal note to the editor: I find the reporting summary etc not useful and they create a lot of work for authors and give the impression of a degree of experimental precision that is not really attainable in most situations (certainly not in my own group).

Author Rebuttals to Initial Comments:

Point-to-point response to the reviewers' comments

We thank the reviewers for their positive and constructive feedback, which aided us to further improve the manuscript. Below is a point-by-point response to all comments and a detailed description of all changes we have made to our manuscript after considering their suggestions. The changes are highlighted in yellow in the revised manuscript.

Reviewer #1

[1.1] The authors use well-established nucleotide ATP-like analogues to infer the ATP F-actin structural state. However, at the resolutions at which the authors are working and at the level of individual water molecules, can they really be certain that these analogues are truly structurally analogous at atomic resolution? It would be very exciting and novel to see the authors apply their processing methods to dynamic actin filaments, enabling them to capture physiologically relevant nucleotide-dependent structural transitions that do not rely on analogues.

Reply 1.1 (see also **Reply 2.11**): The BeF_3^- group, which represents the γ -phosphate mimic in our studies, cannot be considered “truly analogous” of phosphate as it is comprised of different atoms. However, as mentioned in the manuscript, the orientation and bond lengths of ADP-BeF_3^- in our F-actin structures are highly similar to those of ATP in X-ray structures of G-actin. Thus, BeF_3^- represents a suitable γ -phosphate analog for elucidating structures of F-actin in a state that resembles the ATP ground state. Importantly, ADP-BeF_3^- has been characterized as an ATP ground state analog based on high-resolution structures of a wide range of molecular machines, such as maltose transporters (see Oldham and Chen, PMID: 21825153) and sarcoplasmic Ca^{2+} -ATPases (see Møller *et al*, PMID: 20809990).

We fully agree with the reviewer that it would be very exciting to capture a true pre-hydrolysis state of F-actin. However, actin hydrolyzes ATP within seconds of polymerization (rate 0.3 s^{-1}). Therefore, to obtain a cryo-EM sample in which the majority of filaments are in the ATP state, one would have to vitrify actin on a cryo-EM grid within a second of polymerization. This would require fast mixing of G-actin-ATP with a high-salt buffer to reach 100 mM KCl and 2 mM $\text{CaCl}_2/\text{MgCl}_2$ final concentration and spraying onto a grid. Unfortunately, this is unattainable with our standard vitrification protocols using a Vitrobot Mark IV. Instead, this time-resolved cryo-EM approach would rather require a specialized mixing/spraying device. There are only a few examples worldwide where these devices have been successfully used by specialized labs after years of fine-tuning and optimization. Therefore, such an approach is not feasible within our current study but on our list of future investigations.

[1.2] The authors state (p10) that their data show that the Ca^{2+} ion is more mobile within the actin catalytic site and that this could decrease filament stability and explain these filaments' higher depolymerization rate. It is not clear how reduced stability arises from ion mobility. Does this reduced stability manifest in their structures? If not, why not?

Reply 1.2: Thank you for pointing this out. In the nucleotide-binding site of the Ca^{2+} -ADP-bound F-actin structure, we do not observe lower local resolution, less resolved sidechains and higher B-factors of amino acids compared to the other structures. Thus, there is no structural evidence for a reduced stability. We have therefore removed the sentence: “This difference at the active site probably decreases the stability of the filament and could explain the higher depolymerization rates of Ca^{2+} -actin compared to Mg^{2+} -actin¹⁸” from the manuscript. The higher depolymerization rates of Ca^{2+} -F-actin may depend on differences that we cannot observe in our structures, such as for example different conformations at the filament ends.

[1.3] As depicted in Figure 5, evidence for the actin subunit back door by which P_i is released is inferred to be formed by the side chains of R177 and N111 and the backbones of methylated histidine 73 (H73) and G74, and is in a closed conformation in both ADPP_i and ADP states of Mg²⁺-F-actin and Ca²⁺ of F-actin. Can more extensive EM data processing in this area provide more direct evidence for conformational change in this region? Could the authors include molecular dynamics experiments to provide evidence about the behaviour of the back door?

Reply 1.3 (see also **Reply 2.15**): The evidence that the P_i -release back door is formed by R177, N111, H73 and G74 is largely based on a pioneering molecular dynamics study by Wriggers and Schulten from 1999 (PMID: 10223297). This study was based on a ATP-G-actin-gelsolin segment-1 complex. Our high-resolution structures of F-actin provide no evidence for any flexibility within the region of the proposed back door, nor in any other potential back doors. Because no flexibility is observed and the density is of high quality, we believe that more extensive processing will not yield additional insights.

Our observations strongly indicate that P_i release occurs in a transient, high-energy state of F-actin, which is a new finding by itself. We indeed think that state-of-the-art MD simulations could provide new insights into the path of P_i release. However, as P_i release occurs in the range of minutes (rate 0.006 s⁻¹), such simulations would require extensive optimization and CPU time and hence represent a completely new project that could easily take more than a year of work. Including MD simulations was therefore not within the scope of the current study. In fact, we would like to emphasize that our work highlights that P_i release from the F-actin interior is far from completely understood, contrary to statements from many articles that consider the release mechanism through the R177-back door as solved. Thus, the mechanism of P_i release could represent a prevalent theme of future research within the actin field.

[1.4] In figure 6, several rearrangements of the D-loop and the C-terminus at the intrastrand interface in the actin filament, in the presence of Ca²⁺ and Mg²⁺, are described, but more information is needed about the methodology used during the focused classification steps. Specifically, the authors should clearly explain why and how they used two initial model in order to classify the particles. In particular, how did they mitigate the possibility of model bias arising from the use of high value of T (500) combined with a high-resolution filtering of initial models (4 Å). Could alternative image processing strategies – e.g. with the use of signal subtraction combined with mask classification and refinement – provide additional validation of the structural interpretations arising from these analyses?

Reply 1.4: The reviewer highlights an important point. We have described our classification approach now in more detail within the methods section of the revised manuscript.

Firstly, the D-loop is a very tiny region of ~10 amino-acid residues. It is expected that for such a small region within the map (and hence a very small mask compared to the full box), more weight needs to be put on the experimental data, resulting in a higher regularization parameter T. This has been discussed in posts on the ccpem mailing list, see for example <https://www.jiscmail.ac.uk/cgi-bin/wa-jisc.exe?A2=ind1909&L=CCPEM&P=R114857>. We updated the manuscript accordingly: “After optimization of the tau2fudge parameter, which required a high value due to the small size of the mask compared to the full box, this classification ...”.

We have also attempted to use signal subtraction, but that did not yield in improved classification results. Class3D with a high T value was performed without image alignment to exclude strong bias during the classification. It is also clear from the data that we have a mix of ‘closed’ and ‘open’ D-loop conformations. Thus, although we require classification with two references to separate the particles in two conformations, we do not look for conformations that are not there. In the subsequent refinement (in which image alignment is performed again) was performed with a reference of the pre-classification map. However, we understand the concern of Reviewer #1 and have now refined the “closed” and “open” particle sets with an initial model low-pass filtered to 8 Å, in which the D-loop conformation is

indistinguishable. The manuscript now states: "... with mixed D-loop conformation (filtered to 8.0 Å, at which the D-loop conformation is indistinguishable) as reference ...". These refinements yielded the same results as our previous study: the intra-strand interface for the central subunit is separated, but all other intra-strand interfaces within the map (that were not used in the masked classification) display a mixed conformation.

[1.5] The results of the cofilin cosedimentation assay in the presence of Ca²⁺-F-actin are used to derive a new nucleotide-dependent recognition mechanism for cofilin binding. The relevant text (p12-13) only refers to Ca²⁺ data but Mg²⁺ data are also presented in ED16/17 figure and should be incorporated into the description. Only a single concentration of cofilin is used in these experiments – do the observations hold up when a more complete titration is performed? This is important given the cooperative nature of cofilin binding to F-actin.

Reply 1.5: Thank you for highlighting this point. The original manuscript indeed did not refer to Mg²⁺-F-actin data, this has now been adjusted in the revision: "To assess this, we incubated cofilin-1 and Mg²⁺- or Ca²⁺-bound F-actin in three nucleotide states and measured cofilin-dependent filament severing."

To validate our observations that the D-loop conformation does not represent the only sensor for cofilin binding and severing, we have, based on the reviewer's suggestions, now repeated the experiment using a broad range of cofilin concentrations (5, 10 and 20 μM). To highlight that the function of cofilin is F-actin severing, rather than just binding, we calculated the cofilin-dependent F-actin severing based on three independent experiments. These experiments reveal that, over the entire concentration range tested, cofilin-1 only efficiently severs both Mg²⁺- and Ca²⁺-F-actin in the ADP state, and not in the other nucleotide states. This supports our statement that cofilin senses the γ-phosphate moiety in F-actin.

Reviewer #2

[2.1] The paper is well written but will need to be re-formatted for Nature. I would suggest presenting hypothetical mechanisms in more cautious language throughout. Also, the number of main figures needs to be reduced, since too much detail is presented in my opinion. The authors might want to remove some of the data on calcium-bound actin from the main figures, for example to increase brevity and clarity.

Reply 2.1: We are grateful for the feedback from Reviewer #2. Also based on editorial comments, we have shortened the manuscript, reduced the number of main figures and have moved some of the Ca²⁺-F-actin figures to the Extended Data.

[2.2] p1: Title: "aging" a good word, seems actin specific? "ATP hydrolysis" instead?

Reply 2.2: We appreciate the suggestion from Reviewer #2. However, changing "aging" to "ATP hydrolysis" would yield a title that does not fully cover the contents of our manuscript; for instance, we also present insights into the ADP-bound state. Therefore, we prefer to keep "aging" in the title.

[2.3] intro: Mention the other subdomain nomenclature at least once (Ia, Ib, IIa, IIb)?

Reply 2.3: We added the following statement in the legend of Figure 1 to introduce the other nomenclature: "Actin subdomains (SD1-4, also known as Ia, Ib, IIa and IIb) are annotated in the central subunit."

[2.4] general: Is looking at calcium-bound actin interesting? Given that not normally bound in cells (intro, p3). It is of course interesting in terms of understanding the mechanism better, but it might need to be phrased as such.

Reply 2.4: Reviewer #2 is correct that Ca^{2+} probably only marginally binds the actin-nucleotide *in vivo*. In the revised manuscript, we therefore focus more on Mg^{2+} -actin and have reduced the part of Ca^{2+} -actin. Nevertheless, we believe that a molecular understanding of Ca^{2+} -actin polymerization is highly relevant for the field, in addition to scientists interested in mechanisms of enzymes. Ca^{2+} has been standardly used in actin purifications, many *in vitro* studies and most G-actin crystal structures over the last 80 years. The reason for this is, that Ca^{2+} -ATP-bound G-actin exhibits slower polymerization kinetics and a higher critical concentration of polymerization. However, what causes the slow polymerization rates of Ca^{2+} -actin has so far remained unknown and our structures provide the molecular explanation for this phenomenon. We have rephrased the respective paragraph in the introduction to make this aspect clearer to the reader.

[2.5] general: Intra-strand contact is sometimes also called longitudinal.

Reply 2.5: In the first sentence in which the term “intra-strand” is introduced, we added “or longitudinal” between brackets to also make the reader familiar with this definition: “We next examined the nucleotide-state dependent conformational mobility of the D-loop (residues 39–51) and the C-terminus at the intra-strand (or longitudinal) interface in the actin filament^{21,40}”.

[2.6] general: Did you consider/try using a transition-state mimic, such as aluminium fluoride (need to admit that I am not sure this works on actin)?

Reply 2.6: Reviewer #2 is right to assume that AlF_4^- can bind to ADP-F-actin (see e.g., Combeau and Carlier, PMID: 2808407) and that ADP- AlF_4^- bound F-actin is proposed to mimic a transition state. However, our current research focused on elucidating the ground nucleotide states in detail. The extensive biochemical and structural characterization of a potential transition state was therefore not within the scope of the current study, but we definitely will consider it for follow-up work.

[2.7] p4: “All three functional states”: is it not possible that there are more functional states that we can't trap? For example, the state where Pi leaves the active site? (see below, p10, missing open “back door” state).

Reply 2.7: We agree that F-actin is capable of adopting functional states that cannot be trapped with current experimental methodology, and that the “All three functional states” statement is misleading. We have changed the sentence to: “Here, we present six ~ 2.2 Å cryo-EM structures of rabbit skeletal α -actin filaments in three functional states, polymerized ...”

[2.8] p6, top: I do not find it “surprising” that there are no extra binding sites for Mg and Pi.

Reply 2.8: We initially wrote “surprisingly” because we do not observe extra binding sites for Mg and Pi, even though these binding sites were previously predicted. Nevertheless, we now have removed the term “surprisingly” from the text.

[2.9] p6: How can we be sure that F-actin is bound to calcium and not a mix of magnesium and calcium (also: methods)?

Reply 2.9: The final step of the purification of actin from rabbit muscle acetone powder is dialysis in “G-buffer” for 2 days, with multiple buffer exchanges. G-buffer contains 0.2 mM CaCl_2 and no Mg^{2+} -salts. Thus, all Mg^{2+} ions that were present in the sample are infinitely diluted and removed. Combined with data that the affinity of Ca^{2+} for ATP-bound G-actin is slightly higher than that of Mg^{2+} and, that exchange is relatively fast (see Estes *et al* PMID: 1527214), it can be concluded that the divalent cation bound to G-actin after the purification is Ca^{2+} . Therefore, actin that was polymerized in the absence of MgCl_2 does not harbor a Mg^{2+} -ion in the active site. To clarify this, we added the following sentence to the methods section, after describing the dialysis: “This ensured that Ca^{2+} was the divalent cation bound in the active site of G-actin.”

[2.10] p8: What are the attack angles of the other two water and their distances (list)?

Reply 2.10: For the Mg^{2+} -F-actin structures, the distance to the Be-atom of the modelled water proposed to represent W_{nuc} is 3.6 Å with an angle of 144°; W_{bridge} is 4.0 Å with an angle of 134°; and the third water is 6.6 Å with an angle of 125°. For the Ca^{2+} -F-actin structures, the distance to the Be-atom of the modelled water proposed to represent W_{nuc} is 3.7 Å with an angle of 137°; W_{bridge} is 4.0 Å with an angle of 125°; and the third water is 6.2 Å with an angle of 127°. We have added a table to Extended Data Fig. 6i in which these distances are now listed.

[2.11] p8 I find the attack angle of W_{nuc} slightly worrying, given the speed of the reaction involved. Could this have to do with the use of BeF_3 , which will cause slightly altered geometry and dimensions? Any way to get to true ATP actin?

Reply 2.11: Although we propose in the manuscript that W_{nuc} may adopt hydrolysis competent and hydrolysis less competent configurations, we indeed cannot exclude that the orientation of the nucleotide is slightly altered when ADP-BeF_3^- is used. We have now added a sentence to the manuscript to underline this: “Although it cannot be excluded that nucleotide orientation is slightly altered between ADP-BeF_3^- -bound and ATP-bound F-actin, inspection ...”. The development of a time-resolved approach for the elucidation of a true ATP state of F-actin was unfortunately not feasible within the current study. We also refer Reviewer #2 to our elaborate **Reply 1.1**.

[2.12] p8: The putative mechanism of water attack and proton transfer is hypothetical at this point I would suggest. Please phrase as such. This applies to many hypothetical scenarios predicted from the structures, more cautious wording is warranted, generally.

Reply 2.12: We agree that the proposed mechanism of ATP hydrolysis is hypothetical, although we believe that our predictions are justified by the experimental data. We have adjusted the manuscript and have introduced more cautious wording (more cautious wording is highlighted in **bold**): “ W_{bridge} **may** represent a Lewis base with a high potential to activate W_{nuc} and **potentially** act as an initial proton acceptor during hydrolysis, followed by transfer of the proton to D154, as previously predicted by simulations^{57,58}, or alternatively, to H161. In conclusion, we **propose** that Q137 coordinates W_{nuc} but that the hydrogen bond network comprising W_{bridge} , D154 and H161 is responsible for the activation of W_{nuc} and proton transfer”.

[2.13] p8: Q137 and Q137/D154/H161 could have been investigated in more details with mutants.

Reply 2.13: We already refer to these mutants in the manuscript: “Indeed, the ATP hydrolysis rates of the Q137 to alanine (Q137A) actin mutant are slower but not abolished³⁴, whereas the triple mutant Q137A/D154A/H161A-actin exhibits no measurable ATPase activity³⁵.” – which refers to previous studies that investigated the effect of alanine mutations of these residues on ATP hydrolysis. Because these mutants have been characterized previously by our group (ref 35: Funk *et al*, PMID 31647411 - for the triple mutant) and others (ref 34: Iwasa *et al*, PMID: 18515362 – for the Q137A mutant), we have not included them in our current study.

[2.14] p9: The putative explanation of slower polymerisation and hydrolysis by calcium actin is quite convincing.

Reply 2.14: Thank you for the positive feedback.

[2.15] p10: It is somewhat puzzling that the P_i “back door” is not visible in the ADP-P_i or ADP F-actin structures. See below, how the ADP-P_i state was generated.

Reply 2.15: See also **Reply 1.3**. The proposed “back door” of P_i release is indeed closed in all our structures, as well as any other potential back door, which let us to propose that P_i release occurs transiently in a high energy state. This hypothesis is further supported by kinetic data. Namely, P_i

remains bound to F-actin in the range of several minutes, whereas its affinity is low (~1.5 mM, see Carlier and Pantaloni, PMID: 3335528). Accordingly, the binding and dissociation of γ -phosphate mimics such as BeF_3^- from ADP-F-actin is, as described in previous studies, “very slow” (see Combeau and Carlier, PMID: 3182855). Thus, phosphate molecules/mimics do not freely diffuse in and out of the F-actin active site, which supports that binding and dissociation occur in a short-lived state that we cannot easily trap with averaging methods such as cryo-EM. Of note: we incubated our samples >6 hours in BeF_3^- or P_i to ensure saturation of the binding site (see methods).

[2.16] p11 & Movie S4: very minor changes are being discussed ... how can we be confident these are real differences, especially given that all atomic model building progressed from one model as far as I understood?

Reply 2.16: We would like to emphasize that our high-resolution structures now, for the first time, allow us to discriminate minor changes because we can confidently model the positions of waters and amino-acid sidechains. All six presented structures are highly similar, it was therefore convenient to use the Ca^{2+} -ADP bound F-actin structure as starting model for modelling. However, in the subsequent elaborate, iterative process of manual modeling in Coot and real-space refinement in phenix, all residues and solvent molecules were refined in the experimental density map of the specific state. The observed differences can therefore be attributed to changes between the different states.

[2.17] p12: Does rabbit actin have all the modifications as human actin (since human cofilin was used)?

Reply 2.17: Rabbit skeletal α -actin indeed harbors the same hallmark post-translational modifications as human skeletal α -actin, such as N-terminal acetylation and the methylation of residue H73. In general, the majority of biochemical studies on the actin-cofilin complex have been performed with rabbit actin, simply because it is the most convenient to purify in large quantities. Thus, the established purification protocol and well-characterized biochemistry also made rabbit actin a prime candidate for our structural studies.

[2.18] p13: Not everything that affects rates is visible in atomic structures. For example, electrostatic interactions can drive affinities very significantly (Ca vs Mg / ATP vs ADP actin). ADP F-actin can have a much lower longitudinal affinity, purely based on changed electrostatics (2 vs 3 phosphates). This is thought to drive de-polymerisation in tubulin-like proteins, for example.

Reply 2.18: This is an excellent suggestion. In the conclusion section of the manuscript, we now refer to a study that shows that the nucleotide state affects the mechanical properties of the filament.

[2.19] p15: I note in the methods that the ADP- P_i state was created by adding P_i to ADP-actin, not by hydrolysis. That state could be different from a state that resulted from hydrolysis of ATP, and this may have shown the P_i “back door” open? I am aware that hydrolysis is much more difficult to control.

Reply 2.19: See also **Replies 1.3 and 2.15**. P_i release occurs in the order of minutes, but it is expected that some P_i molecules, especially those close to the filament ends, will be released faster. Therefore, a protocol aimed to capture ADP- P_i right after polymerization and hydrolysis would effectively result in a mixture between ATP, ADP- P_i and ADP states of F-actin. With the used protocol, we could ensure that the P_i -binding site in F-actin was saturated and that we isolated a full ADP- P_i state. Indeed, the cryo-EM densities of the P_α and P_β of ADP and of P_i are of comparable intensity, indicating full occupancy. It is generally accepted within the actin field that P_i binding is reversible from a kinetic perspective (see e.g., Carlier and Pantaloni, PMID: 3335528 and Fujiwara *et al*, PMID: 17517656). Indeed, previous cryo-EM studies by other groups (see e.g., Chou and Pollard, PMID: 30760599 and 33214556) have used a similar approach of generating the ADP- P_i state. In addition, structures of F-actin copolymerized with cyclic peptide toxins jasplakinolide and phalloidin (see Pospich *et al*, PMID: 32084355) have revealed a P_i -binding site highly similar to that in our current structures (including a closed back door). The toxins strongly inhibit P_i release and therefore, these structures show P_i that is

the product of ATP hydrolysis. Therefore, combined with **Reply 2.14**, we do not believe that a different preparation of the ADP-P_i state would yield an open back door.

[2.20] p17: What particle, CTF and aberration parameters were refined in Bayesian polishing in RELION?

Reply 2.20: Bayesian polishing and contrast transfer function (CTF) refinement refer to two separate image processing steps that improve the quality and resolution of the reconstruction. During cryo-EM data collection, the electron beam induces movements of the protein within the vitreous ice layer. To correct for this beam-induced motion, cryo-EM micrographs are typically collected as ‘movies’ that consist of frames (60 – 80 frames in our experiments). At the start of image processing, these frames are aligned to correct for the motion. During the processing, correction with the contrast-transfer function is essential for obtaining high-resolution reconstructions. The contrast-transfer function is dependent on the used defocus and is initially estimated per micrograph. Bayesian polishing and CTF refinements are used in later stages of processing after an initial good-quality density map is obtained. Bayesian polishing employs a Bayesian approach to estimate improved particle trajectories of beam-induced motion and performs dose-weighting. With CTF refinements, the CTF per particle is estimated, which is a more accurate estimation than “per micrograph”. Namely, the ice thickness in each micrograph is never fully uniform, which will affect the CTF. We furthermore used CTF refinements to estimate aberrations caused by microscope imperfections: beam tilt, 3-fold (trefoil) astigmatism, Cs and 4-fold (tetrafoil) astigmatism and anisotropic magnification. More information can be found in the publications describing Bayesian Polishing (Zivanov *et al*, PMID: 30713699) and CTF refinements (Zivanov *et al*, PMID: 32148853). The methods section has been slightly adjusted to make this clearer: “Within Relion 3.1.0, the particles were subjected to Bayesian polishing⁶⁰ for improved estimation of particle movement trajectories caused by beam-induced motion; and to CTF refinements⁶¹ to estimate per-particle defocus values and to correct for beam tilt, 3-fold (trefoil) astigmatism, Cs and 4-fold (tetrafoil) astigmatism, and anisotropic magnification”.

[2.21] p18: For the water molecules: was chemical plausibility taken into account when deciding on the positions of water molecules? How were waters distinguished from ions bound to the proteins? Some waters are not very round ... Crystallography has developed many tools for this to be done more objectively ...

Reply 2.21: Due to the relative “young age” of high-resolution cryo-EM, tools to verify modelled ligands and waters are not as far developed as those for macromolecular crystallography. Therefore, we indeed have paid much attention to model only solvent molecules that make chemical sense (e.g., hydrogen-bonded to amino-acid residues). The Mg²⁺ and Ca²⁺ ions bound to the nucleotide could be modelled based on their geometry and coordination by the nucleotide. Besides the ions in the nucleotide-binding site, we did not identify any solvent density that we, based on geometry and coordinating residues, could unambiguously attribute to an ion. We therefore modelled all these densities as waters. The manuscript states: “Although earlier studies predicted additional Mg²⁺ and P_i binding sites outside of the F-actin nucleotide-binding pocket^{25,40,41}, we did not find evidence for these secondary ion-binding sites in any of our reconstructions”. Specifically, Scipion *et al* (PMID: 30254171) inventoried all cation binding sites in G-actin crystal structures, and predicted that some of these sites may also be present in F-actin. We specifically inspected these sites in all our structures but did not observe clear density for ions, supporting our decision to only model waters.

[2.22] Figure 1: Very well presented. I would have added stereo versions of these as supplement.

Reply 2.22: Thank you. Unfortunately, the limited number of Extended Data Figures does not allow us to include stereo figures. Instead, in Supplementary Videos 1 and 2, we provide 3D views of the images in Fig. 1. Additionally, the maps and models are deposited to, respectively, the EMDB and PDB, which will allow interested readers to look at the structures in 3D.

[2.23] Figure 2: No bond is shown here for Be to P_β?

Reply 2.23: This was indeed not consistent. We now show the bond between the P_β and Be in Fig. 2 and Extended Data Fig. 6.

[2.24] Figure 4a: Again, what is the distance of W_{bridge} to Be? To me, the attack angle looks better for W_{bridge}? (The atomic model and/or stereo would help the reader to form an opinion).

Reply 2.24: See **Reply 2.10**. Unfortunately, the limited number of Extended Data Figures did not allow us to include stereo figures. However, we provide supplementary videos to allow for a better three-dimensional observation of the structures. Specifically in Supplementary Video 3, a 3D view of the nucleotide-binding site is presented.

[2.25] I think there are too many figures ...

Reply 2.25: We believe that, in order to visually clarify important findings to an unexperienced reader, structural biology research profits from manuscripts with many display items. However, also in order to shorten the manuscript, we have reduced the number of main Figures and merged several Extended Data Figures.

[2.26] As far as I can see, only preliminary PDB validation reports have been provided. I thought submissions required full validation reports and PDB IDs these days to make sure structures have actually been submitted and accepted, before manuscript review?

Reply 2.26: Thank you for pointing this out. We are happy to report that all models and maps have now been submitted and that the validation reports did not reveal any large errors. Full validation reports have been included in the present submission.

[2.27] Line numbers would have been helpful throughout.

Reply 2.27: We have now added line numbers to the manuscript.

Reviewer Reports on the First Revision:

Referees' comments:

Referee #1 (Remarks to the Author):

The manuscript of Oosterheert et al is a high-quality structural study that yields novel insights into the properties of actin filaments. In addition to the authors' response to the reviewers' comments, this manuscript also has to be considered in the context of the work of Reynolds et al, which has recently been deposited on bioRxiv (<https://doi.org/10.1101/2022.06.02.494606>), and which is also now cited by Oosterheert et al. There is certainly reassuring consistency between the 2 studies with respect to fundamental nucleotide-dependence of F-actin structures. However, with its narrower focus on the enzymology of F-actin, the work of Oosterheert et al suffers in comparison to Reynolds et al, who apply more creative approaches to understanding actin structural dynamics and thereby opens up new ways of understanding the mechanobiology of the cytoskeleton.

In their revision, Oosterheert et al have addressed a subset of the previously raised points:

Review point 1.1: While it is positive that the authors see that the future use of rapid spray devices will allow more novel approaches to F-actin cryo-EM sample preparation, it is disappointing that such methodologies have not already been implemented for high resolution structure determination. Kontziampasis et al (2019 PMID 31709058) demonstrated that F-actin could be frozen using a rapid freezing device and that a 5.6 Å reconstruction could be determined from a small dataset and without significant optimisation. It can be anticipated that structure determination of such samples will provide critical insight into the true structure of ATP-actin, as oppose to those stabilised by analogues, and would also shed light on the dynamic structural transitions that occur between subunits within filaments as ATP hydrolysis proceeds.

1.2: The authors have acknowledged that their data do not provide an explanation for the reduced stability of Ca-F-actin. They speculate that this phenomenon could arise from effects at the end of filaments. Do their cryo-EM data already provide evidence in support of this idea? In addition, it would be extremely relevant to apply the approaches described by Reynolds et al relating to filament flexibility to explore this phenomenon further.

1.3: The authors' point about the time required to undertake MD simulations to interrogate phosphate release mechanisms is well taken, although such experiments would greatly enrich the current manuscript. Might ADP.Pi F-actin filaments be washed in some way prior to vitrification to stimulate more concerted phosphate release and increase the likelihood of capturing transient release state(s)? Fundamentally, the authors present absence of evidence concerning the phosphate release mechanism which is worthwhile to flag to the field. It would be helpful if the authors were explicit in the manuscript about why current thinking is not correct and how this open question might be addressed in the future.

1.4: The additions to the manuscript concerning classification strategies for the D-loop provides confidence in the methodology applied. The authors should explicitly add to the Methods text how

they excluded that the use of very high T value resulted in overfitting of the data. They should also explicitly state in the relevant section of the Methods why two references were used for the first iteration of classification – were they absolutely certain there were only 2 confirmations and if so, how?

1.5: The authors have clarified and added more data to their observations concerning cofilin binding. One explanation for these observations could be the authors' hypothesis that cofilin directly senses the F-actin phosphate state, but as mentioned by one of the reviewers ([2.18] p13: "Not everything that affects rates is visible in atomic structures." This is elegantly explored by Reynolds et al - cofilin-1 binding may also be sensing mechanical filament properties, which are not captured in the Oosterheert study.

Referee #2 (Remarks to the Author):

Second round review: Oosterheert et al., Nature ms# 422981:

The authors did an excellent and thorough job in answering questions and making changes. I would be happy for this work to progress to publication without delay. A few more comments:

- I agree that looking at dynamic actin would have been exciting but it is a new and difficult project as pointed out by the authors. Looking at dynamic, ATP-hydrolysing F-actin, the slightly distorted geometry around the BeF moiety might get resolved and the "back door" issue pointed out in the manuscript and by both reviewers might also get resolved, although that is less clear since it is most likely a transition state and might only occur very briefly. I wanted to challenge the authors on this important goal, but as said, I do not think the current advances are diminished by this future option.

- Reply 1.4: if using 3D classifications without particle alignment (essentially just sorting particles into a few classes), very small parts of the map can be masked successfully and the danger of bias introduction is small. We have done this on stretches of 5 residues in tubulins very successfully (to distinguish alpha and beta). Has this been explored?

- It is good to see that the manuscript is now more compact and readable, mostly because of de-emphasising the calcium angle, which I am not so keen on as it is non-physiological (but mechanistically interesting, as pointed out by the authors correctly).

- Wording has been changed to more hypothetical phrases in important places, which is good to see since mechanisms are deduced from snapshots, only, and from the use of analogues.

- Reply 2.20: thank you for the detailed explanation of what was done.

Author Rebuttals to First Revision:

2nd round point-to-point response to the reviewers' and editor's comments

We thank the reviewers for their additional feedback on our revised manuscript. Below is a point-by-point response to all comments and a detailed description of all changes we have made to our manuscript after considering their suggestions. The changes are highlighted in yellow in the new version of the manuscript.

Reviewer #1

The manuscript of Oosterheert et al is a high-quality structural study that yields novel insights into the properties of actin filaments. In addition to the authors' response to the reviewers' comments, this manuscript also has to be considered in the context of the work of Reynolds et al, which has recently been deposited on bioRxiv (<https://doi.org/10.1101/2022.06.02.494606>), and which is also now cited by Oosterheert et al. There is certainly reassuring consistency between the 2 studies with respect to fundamental nucleotide-dependence of F-actin structures. However, with its narrower focus on the enzymology of F-actin, the work of Oosterheert et al suffers in comparison to Reynolds et al, who apply more creative approaches to understanding actin structural dynamics and thereby opens up new ways of understanding the mechanobiology of the cytoskeleton.

[1.1] While it is positive that the authors see that the future use of rapid spray devices will allow more novel approaches to F-actin cryo-EM sample preparation, it is disappointing that such methodologies have not already been implemented for high resolution structure determination. Kontziampasis et al (2019 PMID 31709058) demonstrated that F-actin could be frozen using a rapid freezing device and that a 5.6 Å reconstruction could be determined from a small dataset and without significant optimisation. It can be anticipated that structure determination of such samples will provide critical insight into the true structure of ATP-actin, as oppose to those stabilised by analogues, and would also shed light on the dynamic structural transitions that occur between subunits within filaments as ATP hydrolysis proceeds.

Reply 1.1: We fully agree with Reviewer #1 that a structure of a 'true' ATP-state of F-actin would provide additional insights into the G- to F-actin transition and the subsequent F-actin aging process. The reviewer mentions a pioneering study by Kontziampasis *et al* (2019) that used a mixer in combination with a spraying device and rapid-plunge freezing to solve several structures, including those of pre-assembled thin (actin) filaments. The resolution of F-actin reconstructions in that study, as well as in a follow-up study by the same labs (Klebl *et al*. *Acta Cryst D* 2021. PMID: 34605427) is, to the best of our knowledge, limited to ~5 Å or worse, highlighting that sub-3.5 Å structure determination through such an approach is certainly not trivial, not even for the cryo-EM model protein apoferritin. In fact, Kontziampasis *et al* write about their cryo-EM data:

"Tomographic analysis of the ice shows that it varies between 80 and 125 nm, which is thicker than ideal and limits the resolution of the data. Moreover, Thon ring analysis of all collected micrographs shows the main peak lies between 3 and 5 Å [Fig. S4(b)]. Since the resolution of the apoferritin dataset is not particle-number limited (>800 000 asymmetric units), ice thickness is the main determinant of resolution and we conclude that thinner ice will be required to obtain >3.5 Å."

We therefore would like to emphasize that solving structures at resolutions that allow for solvent visualization (sub-2.5 Å) through time-resolved cryo-EM approaches remains extremely difficult, even for those developing the sample preparation devices themselves. Hence, the structure determination of ATP-bound F-actin at sub-2.5 Å resolution will require the dedicated optimization of a time-resolved cryo-EM approach, which was not within the scope of our study.

[1.2] The authors have acknowledged that their data do not provide an explanation for the reduced stability of Ca-F-actin. They speculate that this phenomenon could arise from effects at the end of filaments. Do their cryo-EM data already provide evidence in support of this idea? In addition, it would be extremely relevant to apply the approaches described by Reynolds et al relating to filament flexibility to explore this phenomenon further.

Reply 1.2: As mentioned in Reply 1.2 of our previous point-by-point response, we do not observe structural evidence for the reduced stability of Ca²⁺-F-actin. Hence, the mentioned example of a different conformation at the filament ends is just speculation. Our current cryo-EM data include images of relatively long actin filaments, with no or very few filament-ends visible in each micrograph (see Extended Data Fig. 1a). We therefore cannot average enough filament-end particles to obtain high-resolution end-structures with our current experimental approach. It is indeed also a possibility that the faster depolymerization rates of Ca²⁺-F-actin may be caused by differences in mechano-stability and rearrangements caused by filament bending. We have now added the following sentence to the manuscript: “The absence of discrete differences in amino-acid conformation between the ADP states of Ca²⁺- and Mg²⁺-F-actin suggests that the faster depolymerization rates of Ca²⁺-F-actin may be caused by differences in long-range filament mechano-stability or conformations at filament ends.” Applying approaches to investigate the bending of Ca²⁺-F-actin will require extensive processing strategies and is therefore not within the scope of the study, nor would it be in line with the focus of our manuscript.

[1.3] The authors’ point about the time required to undertake MD simulations to interrogate phosphate release mechanisms is well taken, although such experiments would greatly enrich the current manuscript. Might ADP.Pi F-actin filaments be washed in some way prior to vitrification to stimulate more concerted phosphate release and increase the likelihood of capturing transient release state(s)? Fundamentally, the authors present absence of evidence concerning the phosphate release mechanism which is worthwhile to flag to the field. It would be helpful if the authors were explicit in the manuscript about why current thinking is not correct and how this open question might be addressed in the future.

Reply 1.3: As written in all versions of our manuscript, our current cryo-EM data in the presence of P_i at concentrations high above the dissociation constant (K_D) do not show evidence for any secondary, transient P_i-binding sites. We appreciate the suggestion to wash the excess of P_i away before vitrification, but it would likely result in a mixture of bound and unbound P_i at the active site, without the guarantee that a transient state will be observed. As a result, such an approach would require extensive optimization and will be difficult to perform in a reproducible manner, but we will definitely explore such experiments in future studies. To emphasize that the P_i release mechanism from the F-actin interior is not understood, we have added the following sentences to the manuscript: “which suggests that larger rearrangements are required for P_i release, indicating that the release mechanism remains incompletely understood. We envision that P_i release could be further explored by MD simulations or time-resolved cryo-EM in future research, guided by our structures as high-quality starting models.” Of note, a similar sentence was present in the original version of the manuscript but was removed during manuscript shortening for the first revision.

[1.4] The additions to the manuscript concerning classification strategies for the D-loop provides confidence in the methodology applied. The authors should explicitly add to the Methods text how they excluded that the use of very high T value resulted in overfitting of the data. They should also explicitly state in the relevant section of the Methods why two references were used for the first iteration of classification – were they absolutely certain there were only 2 confirmations and if so, how?

Reply 1.4: We thank Reviewer #1 for helping us clarify our processing strategy to make it understandable for cryo-EM scientists.

The high tau2fudge value was only used in 3D classification without image alignment to separate particles between the two classes and the resulting density maps were not analyzed further. Instead, the

particles of the two classes were refined separately using a map filtered to 8.0 Å, at which the D-loop conformation is indistinguishable and with tau2fudge=1, which is the default value for refinements. Because high tau2fudge value was only used for particle sorting and not for alignments and subsequent map interpretation, overfitting of the data is prevented. We now explicitly mention this in the Methods section: “These refinements were performed with the default tau2fudge=1 to prevent any overfitting. The high tau2fudge value employed during the classification without image alignment was only used to sort particles and not for further map processing and analysis.”

We performed the classification into two classes because we observed evidence for two conformations in the cryo-EM density map of all 2,228,553 particles, as well as in previous cryo-EM structures from our group (see e.g., Merino *et al* 2018. PMID: 29867215). However, we cannot exclude that alternative D-loop conformations are present that are adopted by only a marginal number of the particles. Although such marginally adopted conformations are not distinguishable by our approach, we would like to point out that the degree of flexibility and hence difference in conformations is likely limited due to the small size of the D-loop (~10 residues). We added to the Methods section: “A particle separation into two classes with two initial references was chosen because the two conformations of closed and open D-loop were visible in the refined, non-sharpened reconstruction computed through all 2,228,553 good particles. Potential alternative conformations of the D-loop adopted by only a marginal number of F-actin particles are not distinguishable by our classification strategy.”

[1.5] The authors have clarified and added more data to their observations concerning cofilin binding. One explanation for these observations could be the authors’ hypothesis that cofilin directly senses the F-actin phosphate state, but as mentioned by one of the reviewers ([2.18] p13: “Not everything that affects rates is visible in atomic structures.” This is elegantly explored by Reynolds *et al* - cofilin-1 binding may also be sensing mechanical filament properties, which are not captured in the Oosterheert study.

Reply 1.5: Thank you for pointing this out. Our data indeed support a mechanism where cofilin does not use discrete conformational differences at e.g., the C-terminus and D-loop as only sensor for the nucleotide state, although it cannot be excluded that conformations at the F-actin periphery still play a role. We adjusted a sentence in the manuscript to elaborate on how cofilin may sense the presence of the γ -phosphate moiety: “Our results therefore support the previously proposed model that cofilin cannot form a strong complex with F-actin when the γ -phosphate moiety is present²³ and that it potentially senses the mechanical properties of the filament.”

Reviewer #2

The authors did an excellent and thorough job in answering questions and making changes. I would be happy for this work to progress to publication without delay. A few more comments:

[2.1] I agree that looking at dynamic actin would have been exciting but it is a new and difficult project as pointed out by the authors. Looking at dynamic, ATP-hydrolysing F-actin, the slightly distorted geometry around the BeF moiety might get resolved and the “back door” issue pointed out in the manuscript and by both reviewers might also get resolved, although that is less clear since it is most likely a transition state and might only occur very briefly. I wanted to challenge the authors on this important goal, but as said, I do not think the current advances are diminished by this future option.

Reply 2.1 (see also Reply 1.1): Thank you very much. We will indeed attempt to explore the high-resolution structure determination of the ATP-bound state of F-actin and the P_i release mechanism in future research.

[2.2] Reply 1.4: if using 3D classifications without particle alignment (essentially just sorting particles into a few classes), very small parts of the map can be masked successfully and the danger of bias introduction is small. We have done this on stretches of 5 residues in tubulins very successfully (to distinguish alpha and beta). Has this been explored?

Reply 2.2: (See also Reply 1.4): We have optimized the separation of closed and open D-loop conformations through 3D classification without image alignment. The separation in two classes required two references, because the classification with one reference into multiple classes would otherwise always yield a result where all particles would condense into a single class.

[2.3] It is good to see that the manuscript is now more compact and readable, mostly because of de-emphasising the calcium angle, which I am not so keen on as it is non-physiological (but mechanistically interesting, as pointed out by the authors correctly).

[2.4] Wording has been changed to more hypothetical phrases in important places, which is good to see since mechanisms are deduced from snapshots, only, and from the use of analogues.

[2.5] Reply 2.20: thank you for the detailed explanation of what was done.

Reply 2.3-2.5: Thank you very much for the positive feedback. It greatly helped us to improve and shorten our manuscript.